# VARSHAP: A Variance-Based Solution to the Global Dependency Problem in Shapley Feature Attribution

## Abstract

Feature attribution methods based on Shapley values, such as the popular SHAP framework, are built on strong axiomatic foundations but suffer from a critical, previously underappreciated flaw: global dependence. As recent impossibility theorems demonstrate, this vulnerability is not merely an estimation issue but a fundamental one. The feature attributions for a local instance can be arbitrarily manipulated by modifying the model's behavior in regions of the feature space far from that instance, rendering the resulting Shapley values semantically unstable and potentially misleading.

This paper introduces VARSHAP, a novel feature attribution method that directly solves this problem. We argue that the source of the flaw is the characteristic function used in the Shapley game — the model's output itself. VARSHAP redefines this game by using the reduction of local prediction variance as the characteristic function. By doing so, our method is, by construction, independent of the model's global behavior and provides a truly local explanation. VARSHAP retains the desirable axiomatic properties of the Shapley framework while ensuring that the resulting attributions are robust and faithful to the model's local decision landscape. Experiments on synthetic and real-world datasets confirm our theoretical claims, showing that VARSHAP provides stable explanations under global data shifts where standard methods fail and demonstrates superior performance, particularly in robustness and complexity metrics.

## 1 Introduction

**Feature attribution**  As machine learning models are deployed in high-stakes domains like healthcare and finance, Explainable AI (XAI) is crucial for ensuring fairness, accountability, and trust. A key tool in XAI is feature attribution, which quantifies the contribution of each input feature to a model's prediction (Holzinger et al., 2020). These methods can be global, offering a high-level overview of a model's average behavior Covert et al., 2020, or local, explaining a single, individual prediction. This work focuses on local feature attribution, which is essential for revealing the instance-specific nuances of a model's decision-making process that are often obscured by global perspectives (Guidotti et al., 2018; Murdoch et al., 2019).

Shapley value theory (Shapley, 1953), originating from cooperative game theory, provides a principled method of allocating payout among players. Štrumbelj & Kononenko (2011) suggested that the input features in machine learning models can be treated as players and the model output can be seen as a payout, providing a way to use Shapley value in feature attribution. The strong mathematical underpinnings of Shapley values, characterized by desirable axioms like efficiency, symmetry, null-player, and additivity, have made this framework a popular choice for generating feature attributions (Li et al., 2024). Among these, SHAP (SHapley Additive exPlanations) (Lundberg & Lee, 2017) is a prominent unified method. SHAP assigns each feature an importance value for a particular instance, quantifying its contribution to shifting the model's output from a baseline to its current value. A key strength of the SHAP framework is its model-agnostic nature, allowing it to be applied to any machine learning model without requiring access to its internal structure.

**Inherent flaw of Shapley-based feature attributions** While prized for their axiomatic rigor, Shapley-based feature attribution methods harbor a fundamental vulnerability that undermines their reliability: global dependence. This is not a minor flaw in estimation but an intrinsic and provable weakness of the entire theoretical framework. The issue arises because these methods, in order to satisfy properties like completeness, must incorporate the model's behavior across a global data distribution, even when explaining a single, local prediction.

This vulnerability was definitively formalized in a recent impossibility theorem Bilodeau et al. (2024), which proves that any feature attribution method that is both complete and linear, a class that includes SHAP Lundberg & Lee (2017) and Integrated Gradients Sundararajan et al., 2017, can be arbitrarily manipulated. By altering a model's behavior in regions far from the point of interest, one can produce profoundly different attributions even if the model's local behavior remains unchanged, rendering these methods no better than random guessing for tasks like algorithmic recourse. This is not a flaw in a specific implementation but a foundational problem with the Shapley value concept itself when applied to model explanations. The critical consequence is that the attributions lack stable semantic meaning, as they reflect artifacts of the global model landscape rather than true local sensitivities. For example, a positive SHAP value does not guarantee that increasing a feature will increase the model's output, potentially leading to misleading conclusions for subgroups whose behavior deviates from the population average.

As demonstrated by Bilodeau et al. (2024), marginal SHAP attributions can be manipulated to produce any desired vector while keeping the model's local behavior completely unchanged. In Appendix A.1, we extend this result to show that conditional SHAP suffers from similar issues: Under very mild assumptions[1], the relative importance of any two features — as determined by their attribution values — can be arbitrarily manipulated, regardless of the model's underlying behavior. This fundamental problem affects all approaches that target more efficient or accurate SHAP estimation methods such as KernelSHAP (Lundberg & Lee, 2017), TreeSHAP (Lundberg et al., 2018), on-manifold Shapley(Frye et al., 2020a), or (Aas et al., 2021; Chen et al., 2023; Ketenci et al., 2024), as well as recent work integrating causal reasoning with Shapley explanations (Heskes et al., 2020; Frye et al., 2020b; Jung et al., 2022), since they all rely on the model's output value as the characteristic function. While Janzing et al. (2024) explores variance-based explanations in causal settings, their approach requires known causal graphs, is not model-agnostic, and does not address the global dependency problem that undermines explanation reliability.

**Key contributions** The impossibility theorem reveals that traditional Shapley-based explanations face a fundamental barrier, not a minor flaw. They are intrinsically vulnerable to arbitrary manipulation due to their reliance on a model's global behavior. Overcoming this requires not an incremental fix, but a paradigm shift. This paper introduces that new paradigm. We argue the vulnerability lies in using the model's direct output as the characteristic function in the Shapley game. Our work circumvents this impossibility by fundamentally redefining the game itself. Our key contributions are as follows:

- We theoretically justify that for a local attribution method to be truly local and satisfy intuitive axioms like shift-invariance, its characteristic function must measure the dispersion of model outputs (Appendix A.2).

- We prove that variance is the unique function that satisfies these properties, providing a principled foundation for our approach (Appendix A.3).

- We prove that the global dependence problem persists in Shapley methods using conditional probability distribution in characteristic function (Appendix A.1).

- We introduce VARSHAP, a novel framework for local feature attribution. By using the reduction of local output variance as its core importance metric, VARSHAP is, by construction, immune to the global dependencies that render traditional methods semantically unstable (Section 2.3).

- We provide a comprehensive empirical evaluation on synthetic and real-world datasets. The results confirm that VARSHAP's theoretical advantages translate into superior practical performance (Section 3).

---

[1]We require that the model is piecewise linear and the probability distribution is non-degenerated, the assumptions fulfilled by vast majority of models and datasets.

## 2 VARSHAP: VARiance-based SHapley Additive exPlanations

The global dependency problem inherent in traditional Shapley-based methods stems directly from the choice of the characteristic function, which is the model's raw output. To construct an attribution method that is truly local, we must design a new characteristic function that is, by construction, insensitive to the model's global behavior. The foundational property for such a function is *shift invariance*: a local explanation should not change if the model's outputs are all shifted by a constant value, as this does not alter the relative importance of features to the model's local behavior.

This intuitive requirement leads to a powerful and unique theoretical conclusion. We formally demonstrate in Proposition 1 that any characteristic function designed to be shift-invariant must measure the dispersion of model outputs around their local mean. Furthermore, when combined with a few simple, desirable axioms, such as zero property, sign independence, and additivity, the only function that satisfies these constraints is the squared deviation from the mean. This result provides a rigorous, first-principles justification for using variance as the basis for a new, truly local, and axiomatically sound feature attribution method.

### 2.1 VARIANCE AS THE BASIS FOR FEATURE ATTRIBUTIONS

Traditional SHAP calculates feature importance by measuring how each feature affects the expected model output. Specifically, for a feature $j$ and a subset of features $S$ not containing $j$, SHAP computes the expected model output $\mathbb{E}[\Omega(x_S, X_{-S})]$ where features in $S$ are fixed to their values from the instance being explained, and out-of-coalition features ($X_{-S}$) are treated as random variables sampled from a global background distribution. The difference in expected output when adding feature $j$ to coalition $S$ represents its marginal contribution, which is then weighted by the Shapley kernel across all possible coalitions.

The core methodological innovation of VARSHAP is to redefine local feature importance by quantifying how much the knowledge of a specific feature's value reduces the uncertainty in the model's output within the local vicinity of the instance being explained. The key insight of VARSHAP is to measure feature importance through the lens of variance reduction. Intuitively, when a feature's value is known and fixed (conceptually, when it is added to a coalition of features whose values are already known for the instance in question), the range of possible model outputs, and thus its variance, should decrease if that feature is influential for the prediction at that specific point. The more a feature contributes to reducing this local output variance, the more important it is considered for that particular prediction. Specifically, for each possible subset of features $S$ that does not contain the feature of interest, $j$, VARSHAP compares two scenarios. First, it considers the variance of the model's output when the features in subset $S$ are fixed to their values from the instance being explained, while all other features (including feature $j$ and those not in $S \cup \{j\}$) are perturbed locally around the instance. Second, it considers the variance of the model's output when the features in $S$ and feature $j$ are fixed to their values from the instance, and only the remaining features (those not in $S \cup \{j\}$) are subjected to local perturbation. The difference between these two variances, which represents the reduction in output variance attributable to fixing feature $j$ given the context of known features $S$, is then weighted according to the standard Shapley kernel. Summing these weighted variance reductions across all possible coalitions $S$ provides the VARSHAP attribution value for feature $j$. This approach anchors the explanation in the local behavior of the model, directly addressing the global dependence issue inherent in methods that rely on a global background distribution for feature perturbation or removal.

The choice of variance as the central metric for quantifying feature contributions in VARSHAP is not arbitrary. It stems directly from a fundamental desideratum for feature attribution methods, which is shift invariance. This property is crucial because the absolute scale of model outputs often carries less meaning than the relative contributions of features to those outputs. We would also like for our characteristic function to fulfill the *null player* axiom (i.e. to return zero feature attribution when the variance of model's output is zero, which means that the feature has no influence on the model's output) and to be *symmetric*, to not set preference for increasing or decreasing output value.

This last property requires additional explanation. A prevalent and misleading interpretation of SHAP values is that a positive attribution for a feature directly implies that increasing that feature's value will increase the model's output. This assumption is incorrect because it confuses attribution

with counterfactual prediction. A SHAP value explains how a feature's current value contributed to shifting the prediction from a baseline to its final output. It does not, however, describe how the model will behave if that feature's value is changed. Again, the root of this misunderstanding lies in SHAP's global dependence. A feature can receive a large positive attribution due to the model's behavior on a distant baseline distribution, while in the immediate local neighborhood of the instance, the model might be insensitive or even react negatively to an increase in that feature. Therefore, equating a positive SHAP value with a positive local gradient is a critical error. The attribution explains the "why" of the present prediction, not the "what if" of a future change.

**Proposition 1.** *We propose that any continuous characteristic function $v(x, \Pi, S)$ used for feature attribution that aims to satisfy shift invariance must fundamentally measure the dispersion of model outputs around their local mean. Specifically, such a characteristic function must take the following general form:*

$$v(x, \Pi, S) = \mathbb{E}_{X_{-S} \sim \Pi(x_{-S}|x_S)}[d(\Omega(x_S, X_{-S}) - \mathbb{E}_{X'_{-S} \sim \Pi(x_{-S}|x_S)}[\Omega(x_S, X'_{-S})])]$$

*for some attribution function $d$. We posit that the function $d$ must satisfy the axioms of zero property, sign independence, and additivity to ensure meaningful and consistent attributions. The only function $d$ that satisfies the above axioms, and a simple normalization condition $d(1) = 1$, is $d(x) = x^2$. The formal proof of this proposition is provided in the Appendix A.3.*

Here, $\Omega(x_S, X_{-S})$ is the model's output when features in coalition $S$ are fixed to their values $x_S$ from the instance being explained, and the remaining features $X_{-S}$ are perturbed using $\Pi(x_{-S}|x_S)$ centered around the instance $x$. The inner expectation $\mathbb{E}_{X'_{-S} \sim \Pi(x_{-S}|x_S)}[\Omega(x_S, X'_{-S})]$ represents the local expected model output given that features in $S$ are fixed.

## 2.2 LOCAL PERTURBATION FRAMEWORK

With the justification for using variance as the core of our characteristic function established, the only remaining component to introduce before presenting the full VARSHAP feature attribution formula is the precise mechanism for local perturbation. Given a machine learning model $\Omega : \mathcal{X} \subseteq \mathbb{R}^d \to \mathcal{Y} \subseteq \mathbb{R}$, we define, for any specific instance $x \in \mathcal{X}$ that we wish to explain, a perturbation function $\Pi : \mathbb{R}^d \to \Delta(\mathbb{R}^d)$. Here, $\Delta(\mathbb{R}^d)$ represents the space of all probability distributions over $\mathbb{R}^d$. This perturbation function is designed such that for any given point $x$, $\Pi(x)$ yields a probability distribution with the property that if a random variable $X$ is drawn from this distribution, then its expected value is the point itself, i.e., $\mathbb{E}[X] = x$. This ensures that the feature perturbations, and consequently the analysis of the model's behavior, remain focused on the local neighborhood of the instance, thereby maintaining the locality of the explanation. This process can be interpreted as introducing controlled uncertainty specifically about the values of the out-of-coalition features, allowing us to measure the impact of knowing a specific feature's value by observing how it reduces this local uncertainty.

For a concrete and practical implementation, we propose utilizing a multivariate Gaussian perturbation centered at the instance $x$. Specifically, $\Pi(x)$ is a family of Gaussians defined as $\mathcal{N}(\mu, \Sigma)$, where the mean $\mu$ is set to the instance itself ($\mu = x$), ensuring the perturbation is centered locally. The covariance matrix $\Sigma$ is chosen to be diagonal, meaning $\Sigma_{ij} = 0$ for $i \neq j$. The diagonal elements, representing the variance for each feature's perturbation, are defined as $\Sigma_{ii} = \alpha \cdot \hat{\sigma}_i^2$, where $\hat{\sigma}_i^2$ is the estimated variance of feature $i$ observed in the training dataset, and $\alpha > 0$ is a scaling hyperparameter. This Gaussian perturbation formulation offers several advantages: it aligns well with the theoretical underpinnings (e.g., the additivity property of variance for independent perturbations), naturally accounts for the different scales and typical ranges of various features by using their empirical variances $\hat{\sigma}_i^2$, and is computationally efficient to sample from. Furthermore, the hyperparameter $\alpha$ provides a direct mechanism to control the "locality" of the explanation. Smaller values of $\alpha$ lead to tighter perturbations around $x$, focusing on more immediate local behavior, while larger values explore a slightly broader neighborhood. For efficiency, VARSHAP performs local perturbations using the marginal probability distribution of features, but it also supports conditional probability distributions, a capacity relevant to the lively academic debate concerning the appropriate use of marginal versus conditional probabilities for simulating feature absence Janzing et al. (2020).

## 2.3 VARSHAP FEATURE ATTRIBUTION

Having established the foundational principles of using variance reduction for local feature importance and defined the local perturbation framework, we can now fully define the VARSHAP feature attribution formula. For a given machine learning model $\Omega : \mathcal{X} \subseteq \mathbb{R}^d \to \mathcal{Y} \subseteq \mathbb{R}$, an instance $x \in \mathcal{X}$ to be explained, the set of all feature indices $\mathcal{F} = \{1, \ldots, m\}$, and a local perturbation function $\Pi$, the VARSHAP attribution $\Phi_j$ for a feature $j \in \mathcal{F}$ is defined as:

$$\Phi_j(\Omega, \Pi, x) = \sum_{S \subseteq \mathcal{F} \setminus \{j\}} \omega(|S|) \left( \mathrm{Var}_\Omega(S \cup \{j\}) - \mathrm{Var}_\Omega(S) \right) \tag{1}$$

where $\omega(|S|) = |S|!(k - |S| - 1)!(k!)^{-1}$ is the Shapley kernel, which assigns a weight to each coalition $S$ based on its size $|S|$, out of $k$ total features. Variance of the machine learning model $\Omega$ is formally defined as:

$$\mathrm{Var}_\Omega(S) = \mathbb{E}_{X_{-S} \sim \Pi(x_{-S}|x_S)} \left[ \left( \Omega(x_S, X_{-S}) - \mathbb{E}_{X'_{-S} \sim \Pi(x_{-S}|x_S)}[\Omega(x_S, X'_{-S})] \right)^2 \right] \tag{2}$$

where $x_S$ denotes the components of the instance $x$ corresponding to features in coalition $S$, and $X_{-S}$ (and $X'_{-S}$ for the inner expectation) denotes the random components for features not in $S$, drawn from the local perturbation distribution $\Pi(x_{-S}|x_S)$ which is derived from $\Pi(x)$ by conditioning on $x_S$ (or simply by taking the marginals if perturbations are independent). The term $\Omega(x_S, X_{-S})$ is the model's output with features in $S$ fixed and others perturbed, and $\mathbb{E}_{X'_{-S} \sim \Pi(x_{-S}|x_S)}[\Omega(x_S, X'_{-S})]$ is the local expected output under these conditions. Thus, $\Phi_j$ measures the weighted average marginal contribution of feature $j$ to the reduction of local model output variance across all possible feature coalitions $S$ not containing $j$.

By defining feature attributions using the Shapley value framework applied to the characteristic function $v(S) = \mathrm{Var}_\Omega(S)$ (the local output variance given coalition $S$), VARSHAP inherently satisfies the three fundamental Shapley value axioms, thereby providing theoretically sound and fair explanations. These axioms are:

- *efficiency*: The sum of all VARSHAP attributions, $\sum_{j \in \mathcal{F}} \Phi_j(\Omega, \Pi, x)$, precisely equals the total change in variance from a state where all features are unknown (and thus perturbed according to $\Pi(x)$) to a state where all features are known (fixed to their values in $x$). This total change is $\mathrm{Var}_\Omega(\mathcal{F}) - \mathrm{Var}_{\Omega_f}(\emptyset) = 0 - \mathrm{Var}_{\Omega_f}(\emptyset) = -\mathrm{Var}_{\Omega_f}(\emptyset)$, meaning the sum of attributions equals the negative of the initial total variance under full local perturbation. Thus, the attributions fully account for the overall variance reduction potential.

- *symmetry*: If two distinct features, $j$ and $k$, have an identical impact on the model's output variance for all possible coalitions $S \subseteq \mathcal{F} \setminus \{j, k\}$ (i.e., $\mathrm{Var}_{\Omega_f}(S \cup \{j\}) - \mathrm{Var}_{\Omega_f}(S) = \mathrm{Var}_{\Omega_f}(S \cup \{k\}) - \mathrm{Var}_{\Omega_f}(S)$), then their VARSHAP attributions will be equal, i.e., $\Phi_j = \Phi_k$. Features contributing identically to variance changes receive equal attribution.

- *null player (dummy)*: If a feature $j$ has no influence on the model's output variance regardless of the other features in any coalition (i.e., $\mathrm{Var}_{\Omega_f}(S \cup \{j\}) = \mathrm{Var}_{\Omega_f}(S)$ for all $S \subseteq \mathcal{F} \setminus \{j\}$), then its VARSHAP attribution will be zero: $\Phi_j = 0$. This guarantees that features that do not affect the local output variance receive no importance.

In addition to the additivity axiom of variance, VARSHAP also adheres to the *linearity property*. This property further guarantees the consistency and interpretability of VARSHAP attributions under specific model structures. We define linearity as follows:

**Definition 1.** *A feature attribution method $\Phi$ is linear if for any model $\Omega$ that is decomposable into a sum of single-feature functions, i.e., $\Omega(x) = \sum_{i \in \mathcal{F}} \Omega_i(x_i)$ (where each $\Omega_{f_i} : \mathbb{R} \to \mathcal{Y}$ operates only on feature $x_i$), and for any perturbation function $\Pi$ that generates statistically independent feature distributions for the out-of-coalition features, the attribution for any feature $i \in \mathcal{F}$ at an instance $x \in \mathcal{X}$ is given by: $\Phi(\Omega, \Pi, x)_i = \Phi(\Omega_i, \Pi, x_i)$.*

Here, $\Phi(\Omega_i, \Pi, x_i)$ represents the attribution calculated for the simpler model $\Omega_i$ that solely depends on feature $x_i$. Linearity guarantees that when features contribute independently to the overall model

output (as in an additive model structure), VARSHAP attributions precisely isolate and quantify each feature's individual contribution to the model's output variance. In such scenarios, the importance score assigned to a feature $i$ by VARSHAP for the full model $\Omega$ will be exactly the total variance that feature $i$ would account for if it were the sole input to its respective sub-model $\Omega_i$. A detailed explanation of why the method is linear can be found in Appendix A.4.

The linearity property is important for understanding models with additive structures. For instance, consider a linear regression model $\Omega(x) = \sum_{j \in \mathcal{F}} w_j x_j$. If the features $x_j$ are perturbed independently (e.g., using the proposed Gaussian perturbation with a diagonal covariance matrix), VARSHAP's linearity ensures that the attribution for feature $i$, $\Phi_i(\Omega, \Pi, x)$, will be precisely equal to $-w_i^2 \text{Var}(X_i^\Pi)$. Here, $w_i$ is the coefficient for feature $i$, and $\text{Var}(X_i^\Pi)$ is the variance of the perturbed feature $X_i$ as defined by $\Pi(x_i)$. The magnitude of this attribution, $w_i^2 \text{Var}(X_i^\Pi)$, is exactly the component of the total output variance $\text{Var}_\Omega(\emptyset)$ that is directly attributable to feature $i$'s individual variability and its impact via $w_i$. This aligns perfectly with the intuition that features with larger squared coefficients or those that exhibit greater variability under local perturbation (if influential) should be assigned greater importance in explaining the model's output variance.

**Computational complexity** In practice, VARSHAP requires numerical approximation at two levels. The variance terms can be expressed using integral notation as $\text{Var}_\Omega(S) = \int \Omega(x_S, x_{-S})^2 \pi(x_{-S}|x_S) dx_{-S} - \left( \int \Omega(x_S, x'_{-S}) \pi(x'_{-S}|x_S) dx'_{-S} \right)^2$ where $\pi(x_{-S}|x_S)$ denotes the probability density function of the local perturbation distribution. Computing VARSHAP thus involves both approximating the Shapley value summation over coalitions (as in the usual approach) and evaluating these nested integrals for each variance term. Since these are well-defined integrals, they can be efficiently approximated using standard numerical integration techniques such as Monte Carlo sampling, which are well understood and developed.

## 3 EMPIRICAL EVALUATION

To demonstrate the practical advantages and unique insights offered by VARSHAP, this section presents a comparative analysis with two prominent and widely adopted model-agnostic feature attribution methods: SHAP Lundberg & Lee (2017) and LIME Ribeiro et al. (2016). These methods were specifically selected for comparison due to their established popularity within the XAI community and their core design principle of model-agnosticism. The experimental comparison of VARSHAP with SHAP and LIME is presented in Section 3.3.

To evaluate VARSHAP and compare its explanatory capabilities, a controlled experiment was constructed using synthetic data designed to mimic a simplified healthcare application. This setup allows for a clear understanding of how different attribution methods perform when the underlying data generating process and model behavior can be precisely defined. The experiment considers a scenario with three distinct sub-populations of patients, labeled A, B, and C. For the training phase, these sub-populations consist of 1000, 5000, and 5000 patient samples, respectively, allowing us to investigate how methods perform with varying group sizes and potentially different underlying relationships. The model utilizes two continuous input features, $X_1$ and $X_2$, which represent hypothetical physiological measurements. These features are used to predict a single continuous output quantity, which can be interpreted as a patient's response to a treatment or a risk assessment score.

Two predictive model types were implemented to assess the attribution methods: Neural Network Models (NNM) — standard feedforward networks with an input layer, three hidden layers (50-70-50 neurons), and an output layer — and Ground-Truth Models (GTM). These are not trained models in the conventional sense but direct programmatic instantiations of the known data-generating functions. GTMs provide crucial insights by eliminating model estimation error typically present in trained models, offering a "ground truth" benchmark for explanation fidelity. All input features $(X_1, X_2)$ were normalized across methods and models to ensure fair comparison.

### 3.1 CASE STUDY 1: DEMONSTRATING THE GLOBAL DEPENDENCY FLAW

For the first case study, the data was generated to create specific conditions for testing feature attribution consistency. This involved creating two distinct datasets. In Dataset 1, all patient groups were designed to follow the mathematical relationship $Y = X_1 + 0.2 \cdot X_2$. This means that the outcome

variable Y for every patient, regardless of their group, is determined by the values of their features $X_1$ and $X_2$ according to this specific formula. Dataset 2 introduces a variation. While patient groups A and B still adhered to the same relationship as in Dataset 1 ($Y = X_1 + 0.2 \cdot X_2$), patient group C was defined by a different formula: $Y = X_1 - 0.05 \cdot X_1 \cdot X_2$. This change in the underlying relationship for group C creates a global distribution shift between Dataset 1 and Dataset 2.

The key experimental design element is that despite the global difference between the two datasets (due to group C's altered behavior), the local conditions for a specific point of interest were kept constant. Specifically, for a patient located at the center of cluster A, represented by the feature value vector $[0, 0]$, the prediction mechanism yields the same result in both datasets. This setup allows for the investigation of whether local attribution methods provide consistent explanations for this specific point when the broader data distribution changes.

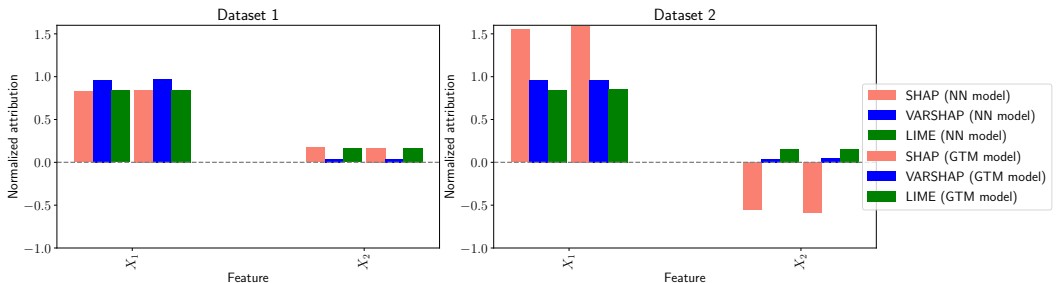

Figure 1: Feature attributions for NNMs and GTMs trained on Dataset 1 (left) and Dataset 2 (right)

The results, illustrated in Figure 1, demonstrate how SHAP, VARSHAP, and LIME perform when analyzing the specific data point $[0, 0]$ in both a neural network and ground-truth models across two distinct datasets. The key observation is that SHAP's attributions for this point varied significantly between Dataset 1 and Dataset 2. In Dataset 1, SHAP assigned positive importance to both features $X_1$ and $X_2$, with $X_1$ receiving a higher attribution value. However, in Dataset 2, SHAP's explanation changed: the attribution for $X_1$ increased, while $X_2$ was assigned a negative attribution value. This occurred despite the fact that the underlying model behavior around the point $[0, 0]$ was designed to be identical in both datasets. In contrast to SHAP, both VARSHAP and LIME provided more consistent attribution values for the point $[0, 0]$ across the two datasets. This stability in explanations from VARSHAP and LIME aligns with the expectation that if the local predictive behavior of a model at a specific point remains unchanged, the explanations for that point should also remain consistent, even if the global data distribution is altered.

It is possible that the observed differences in attribution stem from an artifact specific to neural networks rather than being an issue inherent to SHAP. However, by examining the ground-truth models, which can be considered as having complete insight into the data underlying distribution (akin to an ablation study), we see a strong resemblance in their results (as depicted in Figure 1 ) to those obtained from the NNMs. This close alignment between the GTMs and the NNMs reinforces the argument that the attribution discrepancies are likely due to a fundamental issue within SHAP itself, rather than being a byproduct of the network architecture. SHAP's inherent flaw is its global nature. Changes in distant parts of the data distribution can alter local explanations, meaning they may not accurately reflect the factors driving an individual prediction. VARSHAP avoids this by focusing only on the variance from local perturbations, providing attributions that more faithfully capture the model's immediate behavior.

### 3.2 CASE STUDY 2: CONTRASTING VARSHAP WITH OTHER LOCAL METHODS

In the second case study we specifically investigated scenarios involving non-linear relationships and the inclusion of irrelevant features. This experiment was structured to underscore the limitations of explanation methods that depend on linear approximations and to showcase VARSHAP's resilience in more challenging situations. For this purpose, a synthetic Dataset 3 was generated where the target variable is defined by the non-linear equation $Y = |X_1 + X_2|$. The third feature, $X_3$, was introduced which has no correlation with the target variable, and all features ($X_1, X_2, X_3$) were normalized. The choice of the absolute value function introduces a significant challenge for linear explanation

techniques due to its distinct non-linearity at the point where $X_1 + X_2 = 0$. Furthermore, according to the null player axiom, the attribution for the irrelevant feature $X_3$ should theoretically be zero. Similar to Case Study 1, both GTMs (exactly following the relationship) and NNMs (architecture $[50, 70, 50]$) were implemented. This allows for an analysis of both the theoretical behavior of attribution methods and their practical performance on learned models.

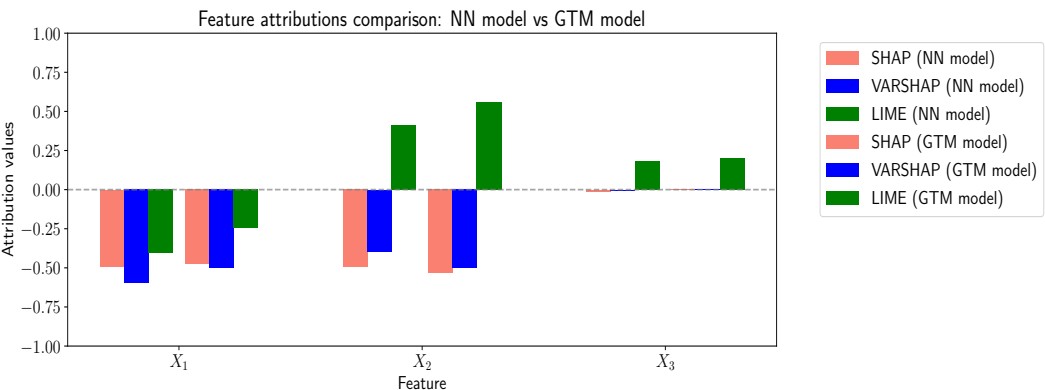

Figure 2: Feature attributions for NNMs and GTMs trained on Dataset 3

In a scenario with strong non-linearity and an irrelevant feature, significant differences between explanation methods emerge, as shown in Figure 2. LIME incorrectly assigns substantial attribution to the irrelevant feature $X_3$, a failure that violates the null player axiom. This error is intrinsic to LIME's methodology, as it also occurs in the ground-truth model (GTM). In contrast, both VARSHAP and SHAP uphold the axiom by correctly assigning near-zero importance to the irrelevant feature. This limitation is further confirmed by a comparison for the GTM in Figure 2 that perfectly implements the absolute value function. Even in this ideal scenario, LIME still incorrectly attributes importance to $X_3$. The core of LIME's issue lies in its fundamental assumption of local linearity, which is ill-suited for handling the sharp non-linearity inherent in the absolute value function. VARSHAP, however, successfully navigates this challenge because its methodology centers on variance reduction. Perturbing the irrelevant feature $X_3$ leads to no change in the output variance, and consequently, VARSHAP correctly assigns it an attribution of zero. To further substantiate the lack of any relationship between $X_3$ and the target variable, partial dependency plots for both models are available in the Appendix A.5, reinforcing the observation that $X_3$ should indeed receive no attribution.

## 3.3 LATEC BENCHMARK RESULTS

Evaluating feature attribution is challenging due to the absence of ground truth and the complex nature of "good" explanations. We address this using a framework based on the LATEC benchmark Klein et al. (2024), applying faithfulness, robustness, and complexity metrics across various models and datasets. Methods are ranked by their median scores per metric over all combinations, providing a robust assessment to systematically compare VARSHAP with SHAP and LIME.

Faithfulness metrics aim to determine if the features highlighted as important by an attribution method genuinely contribute to the model's decision-making process. Robustness metrics, conversely, assess whether features identified as irrelevant are indeed insignificant for the model's predictions. Finally, complexity metrics evaluate the interpretability and conciseness of the explanations provided. Detailed descriptions of all metrics used are provided in Appendix A.6.1.

VARSHAP's performance was + on three datasets with varying characteristics (detailed in Appendix A.6.3). For each dataset, three different model architectures were trained (detailed in Appendix A.6.4). VARSHAP was compared against model-agnostic methods KernelSHAP and LIME with various parameter configurations.

The aggregated rankings (Table 1) show VARSHAP, especially with $\alpha = 0.6$ and $\alpha = 1.0$, and KernelSHAP as the top-performing methods. These approaches significantly outperformed LIME

Table 1: Aggregated ranking of feature attribution methods across all models and metrics

| Method | Avg. Ranking | Method | Avg. Ranking |
|---|---|---|---|
| VARSHAP ($\alpha = 0.6$) | **3.39** | LIME (sparsity = 1.5) | 5.51 |
| VARSHAP ($\alpha = 1.0$) | **3.39** | LIME (sparsity = 5.0) | 5.67 |
| VARSHAP ($\alpha = 0.3$) | 3.64 | LIME (sparsity = 0.5) | 5.81 |
| KernelShap (data sampling) | 3.74 | KernelShap (baseline 0) | 4.86 |

variants, indicating a clear performance advantage. For detailed insights into performance on specific evaluation metrics, refer to Table 2.

Table 2 presents the detailed rankings of methods by individual metrics. The methods show distinct performance patterns. For faithfulness, KernelShap with data sampling often ranks top (e.g., FaithfulnessCorrelation), while VARSHAP is competitive, leading in MonotonicityCorrelation. VARSHAP variants excel in robustness metrics, which indicates VARSHAP's variance-based approach yields explanations more stable to slight input perturbations. Furthermore, VARSHAP leads in complexity metrics, suggesting it produces more concise and interpretable explanations.

Table 2: Rankings by individual metrics (lower is better)

| Method | Faithfulness | | | | | | Robustness | | | Complexity | | |
|---|---|---|---|---|---|---|---|---|---|---|---|---|
| | FC | FC_B | FE | FE_B | MC | MC_B | LLE | MS | RIS | SP | CP | ECP |
| VARSHAP ($\alpha$=0.6) | 2.75 | 5.38 | 2.63 | 4.50 | 3.13 | 4.75 | **2.00** | 2.71 | 5.86 | 2.22 | **1.89** | 3.00 |
| VARSHAP ($\alpha$=0.3) | 2.50 | 4.75 | 2.50 | 5.13 | 3.38 | 5.50 | 2.29 | **1.71** | 6.86 | 2.00 | 2.11 | 5.00 |
| VARSHAP ($\alpha$=1.0) | 2.63 | 5.25 | 2.75 | 5.75 | 2.50 | 5.25 | 2.14 | 3.29 | 5.86 | **1.78** | 2.00 | 2.11 |
| KernelShap (data) | **2.13** | 6.63 | **2.13** | 6.63 | **2.00** | 5.38 | 4.14 | 2.29 | 5.43 | 4.11 | 4.11 | 4.44 |
| KernelShap (0) | 5.63 | 3.75 | 5.63 | 3.88 | 6.50 | 4.00 | 4.43 | 5.00 | **2.86** | 5.00 | 4.89 | 3.22 |
| LIME (1.5) | 6.38 | 3.25 | 6.88 | **3.00** | 6.50 | **3.13** | 7.43 | 7.00 | 2.71 | 7.11 | 7.22 | 4.78 |
| LIME (0.5) | 7.50 | 3.88 | 6.88 | 3.38 | 5.63 | 4.00 | 6.71 | 7.57 | 3.00 | 7.44 | 7.22 | 6.11 |
| LIME (5.0) | 6.50 | **3.13** | 6.63 | 3.75 | 6.38 | 4.00 | 6.86 | 6.43 | 3.43 | 6.33 | 6.56 | 7.33 |

FC: FaithfulnessCorrelation, FC_B: FaithfulnessCorrelation_black, FE: FaithfulnessEstimate, FE_B: FaithfulnessEstimate_black, MC: MonotonicityCorrelation, MC_B: MonotonicityCorrelation_black, LLE: LocalLipschitzEstimate, MS: MaxSensitivity, RIS: RelativeInputStability, SP: Sparseness, CP: Complexity, ECP: EffectiveComplexity

## 4 CONCLUSIONS

The quantitative benchmark evaluation provides strong evidence for VARSHAP's practical advantages, confirming its superior performance against established methods like KernelSHAP and LIME. Crucially, VARSHAP's consistent top-ranking in robustness metrics is not an incidental finding, it is the direct empirical validation of our central thesis. This result demonstrates that VARSHAP's theoretical design, specifically its resilience to the global dependency problem, translates into measurably more stable and reliable explanations in practice. Furthermore, VARSHAP consistently produces more concise and less complex explanations while also exhibiting stability across its different $\alpha$ parameter values, indicating that precise tuning is not critical for achieving strong performance. In practice, even for a much broader range of $\alpha$ values, the method produced well-behaved explanations (assuming the model behaves well in such a neighborhood). Thus, it is up to practitioners and their domain knowledge to define how large a region they want to explain. These findings collectively confirm that our axiomatically-derived, variance-based characteristic function successfully solves a fundamental flaw in prior methods. The result is a feature attribution tool that retains the desirable axiomatic properties of SHAP but delivers explanations that are demonstrably more robust and trustworthy.

The current implementation of VARSHAP relies on a simplified, independent Gaussian perturbation mechanism. A primary direction for future work is to extend this framework to more sophisticated, data-aware distributions, such as conditional probabilities, which could better model complex feature correlations. Such an extension may not only improve attribution for dependent features but could also enhance robustness against adversarial attacks on explanations (Slack et al., 2020), which often exploit out-of-distribution samples to mislead the interpretation method.

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

# A APPENDIX

## A.1 ARBITRARY MANIPULATION OF CONDITIONAL SHAP ATTRIBUTION DIFFERENCES

We prove that conditional SHAP attribution differences between any two features can be made arbitrarily large or small under mild non-degeneracy conditions, while keeping local counterfactual behavior completely fixed. We proof it similar to Bilodeau et al. (2024) for a broad class of piecewise-linear models. This demonstrates that conditional SHAP values can be misleading in the similar way as marginal when comparing feature importance, as the relative attributions depend on model behavior far from the point of interest rather than local counterfactual properties.

**Assumptions**

**Assumption 1.** *[Additive Function Class] Let $f : \mathbb{R}^p \to \mathbb{R}$ be of the form:*

$$f(x) = \sum_{i=1}^{p} f_i(x_i)$$

*where each $f_i : \mathbb{R} \to \mathbb{R}$ is defined as:*

$$f_i(x_i) = \begin{cases} f_i^L(x_i) & \text{if } x_i \in (x_i^L, x_i - \delta_i) \\ g_i(x_i) & \text{if } x_i \in [x_i - \delta_i, x_i + \delta_i] \\ f_i^R(x_i) & \text{if } x_i \in (x_i + \delta_i, x_i^R) \end{cases}$$

The linear components are defined as:

$$f_i^L(x_i) = \beta_i^L \cdot (x_i - x_i + \delta_i) + g_i(x_i - \delta_i) \tag{3}$$

$$f_i^R(x_i) = \beta_i^R \cdot (x_i - x_i - \delta_i) + g_i(x_i + \delta_i) \tag{4}$$

where $\beta_i^L, \beta_i^R \in \mathbb{R}$ are manipulable parameters, and $g_i$ specifies the local counterfactual behavior.

**Assumption 2** (Non-degeneracy Conditions). *For features $i, j$, assume: For features $i, j$, assume there exists probability mass outside the neighborhood for both marginal and conditional distributions: $C_{i,L,\emptyset} \neq 0$ and $C_{i,L,\{j\}} \neq 0$.*

**Compact Notation for Conditional Expectations** For any subset $S \subseteq [p]$, define the conditional distribution $\mu_{i|S}$ of $X_i$ given $X_S = x_S$.

**Definition 2** (Compact Coefficients). *Define:*

$$C_{i,L,S} := \mathbb{E}_{X_i \sim \mu_{i|S}}[X_i \cdot \mathbf{1}\{X_i \in (x_i^L, x_i - \delta_i)\}] - (x_i - \delta_i) \cdot \mu_{i|S}((x_i^L, x_i - \delta_i)) \tag{5}$$

$$C_{i,R,S} := \mathbb{E}_{X_i \sim \mu_{i|S}}[X_i \cdot \mathbf{1}\{X_i \in (x_i + \delta_i, x_i^R)\}] - (x_i + \delta_i) \cdot \mu_{i|S}((x_i + \delta_i, x_i^R)) \tag{6}$$

$$G_{i,S} := G_{i,L,S} + G_{i,R,S} + G_{i,0,S} \tag{7}$$

*where $G_{i,L,S}$, $G_{i,R,S}$, and $G_{i,0,S}$ represent terms involving $g_i$. Lets note that $C_{i,L,S} < 0$ and $C_{i,R,S} > 0$.*

The conditional expectation becomes:

$$\mathbb{E}_{X_i \sim \mu_{i|S}}[f_i(X_i)] = \beta_i^L C_{i,L,S} + \beta_i^R C_{i,R,S} + G_{i,S}$$

**Main Result**

**Theorem 1** (Arbitrary Attribution Difference Manipulation). *For a function class following the construction from Assumption 1 and a distribution fulfilling Assumption 2, for any $p \geq 2$ features and any pair $(i, j)$ with $i \neq j$, the difference $\phi_i^{cond} - \phi_j^{cond}$ can be made arbitrarily large (positive or negative) while keeping the local behavior functions g.*

*Proof.* Without loss of generality, we can assume that the probability mass is non-zero at the $(x_i^L, x_i - \delta_i)$ interval. Setting $\beta_k^R = 0$ for all $k$ and $\beta_j^L = 0$ for transparency, we focus on manipulating $\beta_i^L$.

Using Shapley weights $w(|S|) = \frac{|S|!(p-|S|-1)!}{p!}$, the conditional SHAP attributions are:

$$\phi_i^{cond} = \beta_i^L \left[ \sum_{S \subseteq [p] \setminus \{i,j\}} w(|S|) \cdot (-1) \cdot C_{i,L,S} + \sum_{S \subseteq [p] \setminus \{i,j\}} w(|S|) \cdot (-1) \cdot C_{i,L,S \cup \{j\}} \right] + \text{const}_i$$

(8)

$$\phi_j^{cond} = \beta_i^L \sum_{S \subseteq [p] \setminus \{i,j\}} \left[ w(|S|) \cdot (-1) \cdot C_{i,L,S} + w(|S|+1) \cdot (+1) \cdot C_{i,L,S \cup \{j\}} \right] + \text{const}_j \quad (9)$$

The difference yields:

$$\phi_i^{cond} - \phi_j^{cond} = \beta_i^L \sum_{S \subseteq [p] \setminus \{i,j\}} \left[ w(|S|) \cdot (-1) \cdot C_{i,L,S \cup \{j\}} - w(|S|+1) \cdot (+1) \cdot C_{i,L,S \cup \{j\}} \right] + \text{const}$$

(10)

$$= \beta_i^L \sum_{S \subseteq [p] \setminus \{i,j\}} (w(|S|) + w(|S|+1)) \cdot (-1) \cdot C_{i,L,S \cup \{j\}} + \text{const} \quad (11)$$

Since $C_{i,L,S \cup \{j\}} < 0$ and $w(|S|) + w(|S|+1) > 0$, the coefficient of $\beta_i^L$ is strictly positive and non-zero. Therefore, by choosing $\beta_i^L$ arbitrarily large (positive or negative), we can make $\phi_i^{cond} - \phi_j^{cond}$ arbitrarily large in either direction, while the local behavior offunction remain completely unchanged. □

## A.2 SHIFT INVARIANCE AND CENTERED DISTRIBUTIONS

We begin by establishing that a function of a random variable satisfying shift invariance must be expressible in terms of centered random variables.

**Proposition 2.** *Let $X$ be a random variable representing model outputs, and let $d$ be a distribution functional satisfying shift invariance, i.e., $d(X + c) = d(X)$ for all constants $c$. Then $d$ depends only on the distribution of the centered random variable $X - \mathbb{E}[X]$.*

*Proof.* By shift invariance, for any constant $c$:

$$d(X + c) = d(X) \quad (12)$$

In particular, for $c = -\mathbb{E}[X]$:
$$d(X - \mathbb{E}[X]) = d(X) \quad (13)$$

This shows that $d$ depends only on the distribution of the centered random variable and that without loss of generality, we may assume $\mathbb{E}[X] = 0$ when studying properties of $d$. □

## A.3 DETERMINATION OF THE ATTRIBUTION FUNCTION $d$

In the context of the SHAP framework, we consider expectations over features outside the coalition. Our characteristic function takes the form:

$$v(x, \Pi, S) = \mathbb{E}[d(\Omega(x_S, X_{-S}) - \mathbb{E}[\Omega(x_S, X_{-S})])] \quad (14)$$

Where $d$ is a continuous function we seek to determine. By Proposition 2, we can simplify by assuming $\mathbb{E}[\Omega(x_S, X_{-S})] = 0$ without loss of generality, giving us:

$$v(x, \Pi, S) = \mathbb{E}[d(\Omega(x_S, X_{-S}))] \quad (15)$$

We propose three axioms that the attribution function $d$ should satisfy:

- *zero property*: $d(0) = 0$. This axiom dictates that if perturbing features locally around the instance $x$ (while features in $S$ are fixed) does not lead to any change in the model's output whatsoever (i.e., the deviation from the local expected value is zero), then these perturbed features should collectively receive zero attribution in that specific context.

- *sign independence*: $\forall x \in \mathbb{R} : d(-x) = d(x)$. This property ensures that positive and negative deviations from the local expected value are treated symmetrically. The magnitude of the deviation is what matters for attribution, not its direction. This means the method does not inherently prefer features that increase the model's output over those that decrease it, or vice-versa, when assessing their contribution to output variability.

- *additivity*: For independent random variables $A$ and $B$, representing independent contributions to the overall deviation from the expected value, we require $\mathbb{E}[d(A + B)] = \mathbb{E}[d(A)] + \mathbb{E}[d(B)]$. This property is essential for ensuring that our attribution method behaves additively when combining the effects of independent sources of variation. If the total deviation is a sum of independent parts, their attributed importance should also sum up accordingly.

**Theorem 2.** *The only function $d$ satisfying Axioms 1-3 and normalization $d(1) = 1$ is $d(x) = x^2$.*

*Proof.* To simplify, we may assume $\mathbb{E}[X] = \mathbb{E}[Y] = 0$ for any random variables considered. Let's also assume bounded random variables to ensure existence of all expected values.

By Sign Independence (Axiom 2), $d$ must be an even function. Let's assume $d$ is continuous.

The Additivity axiom requires that for independent random variables $X$ and $Y$ with zero means:

$$\mathbb{E}[d(X + Y)] = \mathbb{E}[d(X)] + \mathbb{E}[d(Y)] \tag{16}$$

By linearity of expectation, this is equivalent to:

$$\mathbb{E}[d(X + Y) - d(X) - d(Y)] = 0 \tag{17}$$

Consider independent random variables $X$ and $Y$ with symmetric two-point distributions:

$$P(X = s) = P(X = -s) = \frac{1}{2} \tag{18}$$

$$P(Y = t) = P(Y = -t) = \frac{1}{2} \tag{19}$$

For these distributions:

$$\mathbb{E}[d(X)] = \frac{d(s) + d(-s)}{2} = d(s) \tag{20}$$

$$\mathbb{E}[d(Y)] = d(t) \tag{21}$$

$$\mathbb{E}[d(X + Y)] = \frac{d(s + t) + d(s - t) + d(-s + t) + d(-s - t)}{4} \tag{22}$$

Since $d$ is even, $d(-s + t) = d(s - t)$ and $d(-s - t) = d(s + t)$. Thus:

$$\mathbb{E}[d(X + Y)] = \frac{d(s + t) + d(s - t)}{2} \tag{23}$$

The additivity condition becomes:

$$\frac{d(s + t) + d(s - t)}{2} - d(s) - d(t) = 0 \tag{24}$$

When $s = t = 0$, we get $d(0) = 0$, confirming Axiom 1.

Setting $s = kt$ for integer $k$, we can show by induction that:

$$d(kt) = k^2 d(t) \tag{25}$$

**Induction Hypothesis**: Assume $f(kt) = k^2 f(t)$ for some integer $k \geq 1$.

**Goal**: Prove $f((k+1)t) = (k+1)^2 f(t)$

From our functional equation with $s = kt$:

$$\frac{f(kt+t) + f(kt-t)}{2} - f(kt) - f(t) = 0$$

Rearranging to isolate $f((k+1)t)$:

$$f((k+1)t) = 2f(kt) + 2f(t) - f(kt-t)$$

Applying the induction hypothesis:

$$f(kt) = k^2 f(t)$$
$$f(kt-t) = f((k-1)t) = (k-1)^2 f(t) \text{ (for } k > 1)$$

For $k = 1$:

$$f(2t) = 2f(t) + 2f(t) - f(0)$$
$$= 4f(t)$$
$$= 2^2 f(t)$$

For $k > 1$:

$$f((k+1)t) = 2k^2 f(t) + 2f(t) - (k-1)^2 f(t)$$
$$= 2k^2 f(t) + 2f(t) - (k^2 - 2k + 1)f(t)$$
$$= 2k^2 f(t) + 2f(t) - k^2 f(t) + 2kf(t) - f(t)$$
$$= k^2 f(t) + 2kf(t) + f(t)$$
$$= (k^2 + 2k + 1)f(t)$$
$$= (k+1)^2 f(t)$$

This proves the induction step, confirming that $f(kt) = k^2 f(t)$ for all positive integers $k$.

For rational numbers $a/b$:

$$d\left(\frac{a}{b}\right) = \frac{a^2}{b^2} d(1) \tag{26}$$

Because:

$$f(t) = \frac{f(kt)}{k^2} \tag{27}$$

$$\tag{28}$$

$$\text{Setting } t = \frac{1}{b} \text{ and } k = b: \tag{29}$$

$$\tag{30}$$

$$f\left(\frac{1}{b}\right) = \frac{f\left(b \cdot \frac{1}{b}\right)}{b^2} \tag{31}$$

$$= \frac{f(1)}{b^2} \tag{32}$$

Since the function is continuous, this extends to all real numbers:

$$d(x) = x^2 d(1) \tag{33}$$

With the normalization condition $d(1) = 1$, we get:

$$d(x) = x^2 \tag{34}$$

This proves that the squared function is the unique solution in this case.

Now we show that for $d(x) = x^2$ these axioms hold for all random variables, concluding that variance is unique function (up to normalization constant):

1. **Zero Property:** $d(0) = 0^2 = 0$. This axiom is trivially satisfied.

2. **Sign Independence:** $d(-x) = (-x)^2 = x^2 = d(x)$. This confirms that $d$ treats positive and negative deviations equally.

3. **Additivity:** For independent random variables $X$ and $Y$:

$$\text{Var}(X + Y) = \mathbb{E}[((X + Y) - \mathbb{E}[X + Y])^2] \tag{35}$$

$$= \mathbb{E}[((X + Y) - (\mu_X + \mu_Y))^2] \tag{36}$$

$$= \mathbb{E}[((X - \mu_X) + (Y - \mu_Y))^2] \tag{37}$$

$$= \mathbb{E}[(X - \mu_X)^2 + 2(X - \mu_X)(Y - \mu_Y) + (Y - \mu_Y)^2] \tag{38}$$

$$= \mathbb{E}[(X - \mu_X)^2] + 2\mathbb{E}[(X - \mu_X)(Y - \mu_Y)] + \mathbb{E}[(Y - \mu_Y)^2] \tag{39}$$

$$= \text{Var}(X) + 2\mathbb{E}[(X - \mu_X)(Y - \mu_Y)] + \text{Var}(Y) \tag{40}$$

$$= \text{Var}(X) + 2\mathbb{E}[X - \mu_X]\mathbb{E}[Y - \mu_Y] + \text{Var}(Y) \quad \text{(by independence)} \tag{41}$$

$$= \text{Var}(X) + 2 \cdot 0 \cdot 0 + \text{Var}(Y) \tag{42}$$

$$= \text{Var}(X) + \text{Var}(Y) \tag{43}$$

$$\square$$

The variance characteristic function emerges naturally from fundamental axioms about feature attribution. When we combine this result with the Shapley value framework, we obtain VARSHAP, a method that maintains the axiomatic properties of Shapley values while focusing on the local behavior of the model through variance reduction.

## A.4 VARSHAP LINEARITY

For a model $\Omega$ that is decomposable into a sum of single-feature functions, i.e., $\Omega(x) = \sum_{i \in F} \Omega_i(x_i)$ (where each $\Omega_i : \mathbb{R} \to Y$ operates only on feature $x_i$), and for a perturbation function $\Pi$ that generates statistically independent feature distributions for the out-of-coalition features, the attribution for any feature $j \in F$ at an instance $x \in X$ is given by:

$$\Phi(\Omega, \Pi, x)_j = \Phi(\Omega_j, \Pi, x_j) = \text{Var}_\Pi(\Omega_j(X_j)) \tag{44}$$

Where $\text{Var}_\Pi(\Omega_j(X_j))$ represents the variance of the output of function $\Omega_j$ when feature $j$ is perturbed according to the perturbation distribution $\Pi(x)$.

Recall that the VARSHAP attribution for feature $j$ is defined as:

$$\Phi_j(\Omega, \Pi, x) = \sum_{S \subseteq F \setminus \{j\}} \omega(|S|)(\text{Var}_\Omega(S) - \text{Var}_\Omega(S \cup \{j\})) \tag{45}$$

Where $\omega(|S|) = |S|!(|F| - |S| - 1)!/|F|!$ is the Shapley kernel, and $\text{Var}_\Omega(S)$ represents the variance of the model's output when features in $S$ are fixed to their values from instance $x$ and features outside $S$ are perturbed according to $\Pi$.

For an additive model $\Omega(x) = \sum_{i \in F} \Omega_i(x_i)$, when features are perturbed independently (as specified by our perturbation function $\Pi$ with diagonal covariance matrix), we can express the variance terms as follows:

$$\text{Var}_\Omega(S) = \mathbb{E}_{X_{-S} \sim \Pi(x_{-S}|x_S)}\left[(\Omega(x_S, X_{-S}) - \mathbb{E}[\Omega(x_S, X_{-S})])^2\right] \tag{46}$$

Due to the additive structure of $\Omega$ and the independence of features, this can be rewritten as:

$$\text{Var}_\Omega(S) = \mathbb{E}\left[\left(\sum_{i \in S} \Omega_i(x_i) + \sum_{l \in F \setminus S} \Omega_l(X_l) - \mathbb{E}\left[\sum_{i \in S} \Omega_i(x_i) + \sum_{l \in F \setminus S} \Omega_l(X_l)\right]\right)^2\right] \tag{47}$$

Since the features in $S$ are fixed, their contribution to the expected value is constant. Therefore:

$$\text{Var}_\Omega(S) = \mathbb{E}\left[\left(\sum_{l \in F \setminus S} \Omega_l(X_l) - \mathbb{E}\left[\sum_{l \in F \setminus S} \Omega_l(X_l)\right]\right)^2\right] \tag{48}$$

For independent random variables, the variance of a sum equals the sum of the variances:

$$\text{Var}_\Omega(S) = \sum_{l \in F \setminus S} \text{Var}(\Omega_l(X_l)) \tag{49}$$

Similarly, for $S \cup \{j\}$:

$$\text{Var}_\Omega(S \cup \{j\}) = \sum_{l \in F \setminus (S \cup \{j\})} \text{Var}(\Omega_l(X_l)) \tag{50}$$

Therefore, the difference in variance is:

$$\text{Var}_\Omega(S) - \text{Var}_\Omega(S \cup \{j\}) = \sum_{l \in F \setminus S} \text{Var}(\Omega_l(X_l)) - \sum_{l \in F \setminus (S \cup \{j\})} \text{Var}(\Omega_l(X_l))$$
$$= \text{Var}(\Omega_j(X_j)) \tag{51}$$

Substituting this into the VARSHAP attribution formula:

$$\Phi_j(\Omega, \Pi, x) = \sum_{S \subseteq F \setminus \{j\}} \omega(|S|) \cdot \text{Var}(\Omega_j(X_j)) \tag{52}$$

Since $\sum_{S \subseteq F \setminus \{j\}} \omega(|S|) = 1$ (a property of the Shapley kernel) and $\text{Var}(\Omega_j(X_j))$ does not depend on $S$, we get:

$$\Phi_j(\Omega, \Pi, x) = \text{Var}(\Omega_j(X_j)) \tag{53}$$

For the specific case of linear regression models where $\Omega(x) = \sum_{j \in F} w_j x_j$, the attribution for feature $j$ becomes:

$$\Phi_j(\Omega, \Pi, x) = w_j^2 \cdot \text{Var}(X_j) \tag{54}$$

Where $\text{Var}(X_j)$ is the variance of feature $j$ under the perturbation distribution $\Pi$. This confirms that when features contribute independently to the model output, VARSHAP attributions precisely isolate and quantify each feature's individual contribution to the model's output variance.

## A.5 PARTIAL DEPENDENCY PLOTS

**Neural Network Model**

**Ground-truth Model**

## A.6 LATEC BENCHMARK DETAILS

### A.6.1 EVALUATION METRICS

For this evaluation, specific metrics were chosen from three categories: Faithfulness (Faithfulness-Correlation Bhatt et al. (2020), FaithfulnessEstimate Nguyen & Martínez (2020) and MonotonicityCorrelation), Robustness (LocalLipschitzEstimate Alvarez Melis & Jaakkola (2018), MaxSensitivity Yeh et al. (2019), and RelativeInputStability Agarwal et al. (2022)), and Complexity (Sparseness Chalasani et al. (2020), Complexity, and EffectiveComplexity Nguyen & Martínez (2020)).

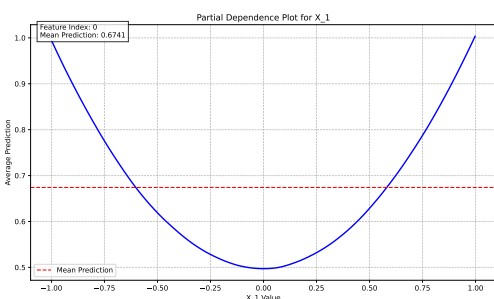
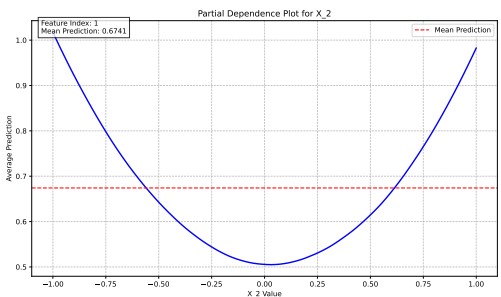

(a) Partial dependency plot for feature $X_1$ for Neural Network Model.

(b) Partial dependency plot for feature $X_2$ for Neural Network Model.

Figure 3: Feature attributions trained on Dataset 3

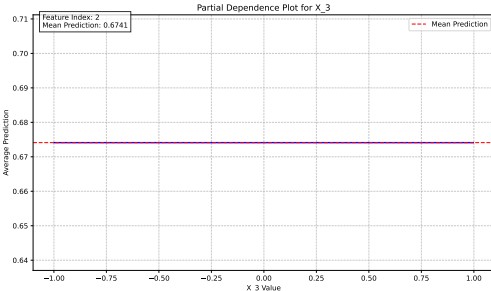

(a) Partial dependency plot for feature $X_3$ for Neural Network Model.

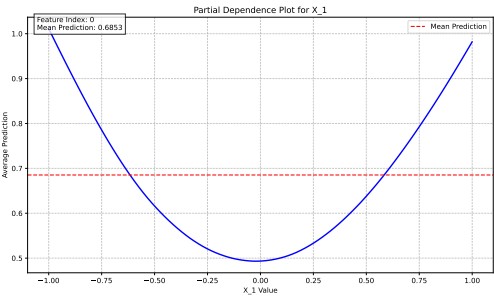
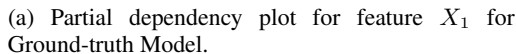

(a) Partial dependency plot for feature $X_1$ for Ground-truth Model.

(b) Partial dependency plot for feature $X_2$ for Ground-truth Model.

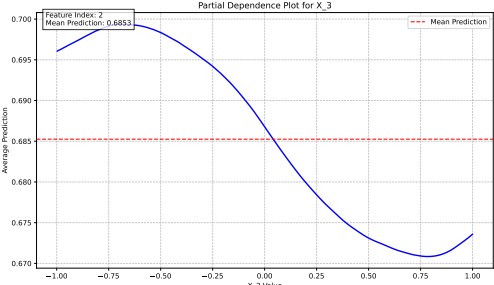

(a) Partial dependency plot for feature $X_3$ for Ground-truth Model.

These particular metrics were selected because they are not inherently image-specific and can be readily adapted to scenarios involving a smaller number of features, rendering them suitable for diverse data modalities. The parameters for each metric were determined based on recommendations from their original publications and the LATEC benchmark, and were subsequently fine-tuned to ensure an appropriate distribution of scores.

### A.6.2 METRICS HYPERPARAMETERS AND HISTOGRAMS

For our evaluation, we configured metrics with the following hyperparameters:

**Faithfulness Metrics**: FaithfulnessCorrelation was configured with 100 runs and subset size 6, using either black" or uniform" perturbation baselines. FaithfulnessEstimate used single-feature steps with both perturbation baseline types. MonotonicityCorrelation used single-feature steps with 10 samples.

**Robustness Metrics**: LocalLipschitzEstimate was implemented with 10 samples, perturbation standard deviation of 0.2, and zero mean. MaxSensitivity used 10 samples with a lower bound of 0.02. RelativeInputStability was configured with 10 samples.

**Complexity Metrics**: Sparseness and Complexity were used with default parameters, while EffectiveComplexity used an epsilon value of 0.05.

All metrics were applied consistently across models and datasets to ensure fair comparison between attribution methods. Below (Figure 7) we present a histogram for the FaithfulnessCorrelation metric on neural network models for the Parkinson dataset. Additional histograms for other metrics, methods, models and datasets can be found in the supplementary materials.

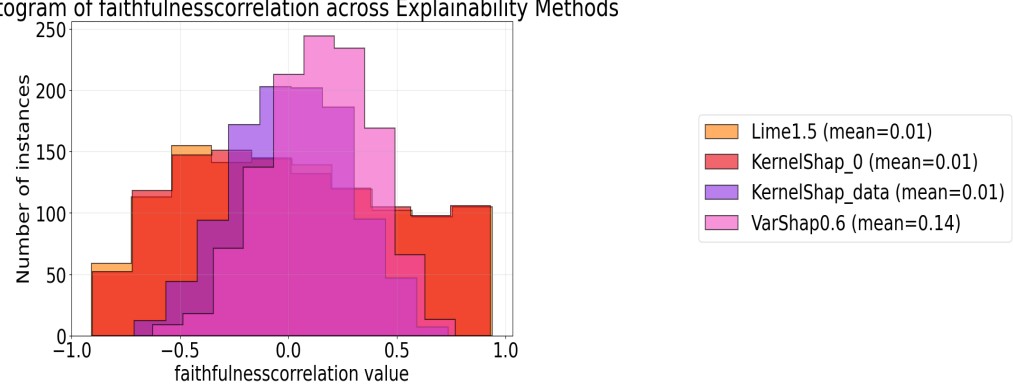

Figure 7: Histogram for metric faithfulnesscorrelation for NN moel for Parkinson dataset

### A.6.3 DATASETS

VARSHAP's performance was evaluated on three datasets:

- **Digits dataset** Pedregosa et al. (2011): 1,797 8x8 handwritten digit images, 64 features
- **Wine Quality dataset** Cortez et al. (2009): tabular wine properties, 11-13 features
- **Parkinsons Telemonitoring** (Tsanas & Little, 2009): with one categorical column removed

### A.6.4 MODEL ARCHITECTURES AND HYPERPARAMETERS

In all cases 80% of dataset was used as a training dataset and 20% as a test dataset. For neural network optimization, we performed grid search hyperparameter tuning over batch sizes (32, 64, 128) and learning rates (0.1, 0.01, 0.001) to identify the configuration yielding optimal model performance. All neural network models used Adam (Kingma, 2014) as an optimizer.

**Digits Dataset Models:**

- **DigitsNNModel:** A simple multi-layer perceptron (MLP) with two hidden layers, each with 128 neurons. The architecture consists of an input layer accepting 64-dimensional feature vectors, followed by two ReLU-activated hidden layers with dropout regularization (rate=0.2), and a final output layer with 10 classes corresponding to digits 0-9. Chosen hyperparameters: **lr=0.01, batch size = 256**

- **DigitsConvNNModel:** A convolutional neural network designed for the 8×8 digits images. The architecture includes a single 2D convolutional layer with 32 filters and 3×3 kernel size, followed by a flattening operation and two fully-connected layers. The hidden layer has 64 neurons with ReLU activation and dropout (rate=0.2), while the output layer has 10 neurons corresponding to the digit classes. Chosen hyperparameters: **lr=0.01, batch size = 256**

- **DigitsTreeModel:** RandomForestClassifier with n_estimators=20, max_depth=4, min_samples_split=2.

**Parkinson Dataset Models:**

- **ParkinsonNNModel:** A standard MLP with two hidden layers, each containing 128 neurons. The architecture begins with an input layer accepting 20-dimensional feature vectors, followed by two ReLU-activated hidden layers with dropout regularization (rate=0.2), and a final single-neuron output layer for regression tasks. Chosen hyperparameters: **lr=0.001, batch size = 64**

- **ParkinsonDeeperNNModel:** A deeper neural network consisting of five layers designed for regression tasks. The architecture begins with an input layer accepting 20-dimensional feature vectors, followed by three hidden layers of 96 neurons each and a fourth hidden layer with 48 neurons, all using ReLU activation and dropout regularization (rate=0.1). The model concludes with a single-neuron output layer for predicting the continuous target variable. Chosen hyperparameters: **lr=0.001, batch size = 32**

- **ParkinsonTreeModel:** RandomForestRegressor with n_estimators=20, max_depth=4, min_samples_split=2.

**Wine Dataset Models:**

- **WineNNModel:** A standard MLP with two hidden layers, each containing 128 neurons. The architecture begins with an input layer accepting 11-dimensional feature vectors (representing wine characteristics), followed by two ReLU-activated hidden layers with dropout regularization (rate=0.2), and a final output layer with 10 neurons. Chosen hyperparameters: **lr=0.01, batch size = 32**

- **WineDeeperNNModel:** A deeper network with five layers designed for the wine quality prediction task. The architecture consists of an input layer accepting 11-dimensional feature vectors, followed by three hidden layers with 96 neurons each and a fourth hidden layer with 48 neurons. All hidden layers use ReLU activation and dropout regularization (rate=0.1). The output layer contains 10 neurons corresponding to wine quality scores. Chosen hyperparameters: **lr=0.001, batch size = 256**

- **WineTreeModel:** RandomForestClassifier with n_estimators=100, max_depth=None, min_samples_split=2.

## A.7 DECISION SURFACES

Figure 8 provides a visual representation of the model's decision-making process for both Dataset 1 and Dataset 2, while Figure 9 depicts decision surface for Dataset 3. Imagine these "decision surfaces" as topographical maps where the contours indicate how the model arrives at a particular prediction. The specific point under examination, [0,0], is marked in red on these maps. What becomes evident from these visualizations is that even though the overall "landscape" or global behavior of the model changes quite dramatically in Dataset 2 (especially in the area corresponding to

group C, which has a different underlying rule), the "local terrain" or gradient immediately surrounding the point [0,0] remains largely the same in both datasets. This means that for data points very close to [0,0], the model makes decisions in a similar fashion in both scenarios. It's worth noting a subtle difference: for the neural network models, the learning process itself introduces very slight alterations to the decision surface even in this local region. In contrast, for the ground-truth models, which are perfectly defined, the decision surfaces in the immediate vicinity of [0,0] are exactly identical. The fact that similar patterns in how importance is attributed to features were seen in both the more complex neural network models and the simpler, perfectly known ground-truth models is significant. It confirms that the way these attribution methods behave (e.g., SHAP's changing attributions versus VARSHAP and LIME's consistency) is a fundamental characteristic of the methods themselves, rather than some quirk or artifact introduced by the model training process.

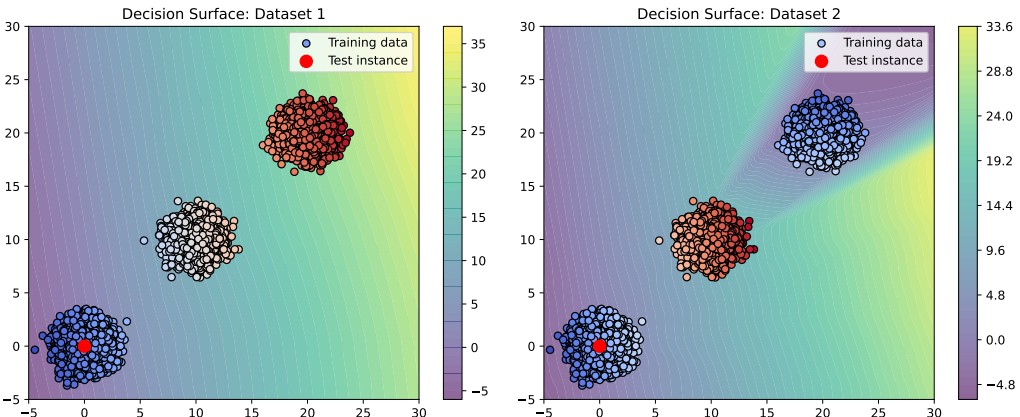

Figure 8: Decision surface visualization of NNMs trained on Dataset 1 (left) and Dataset 2 (right). The test instance at $[0, 0]$ is marked in red.

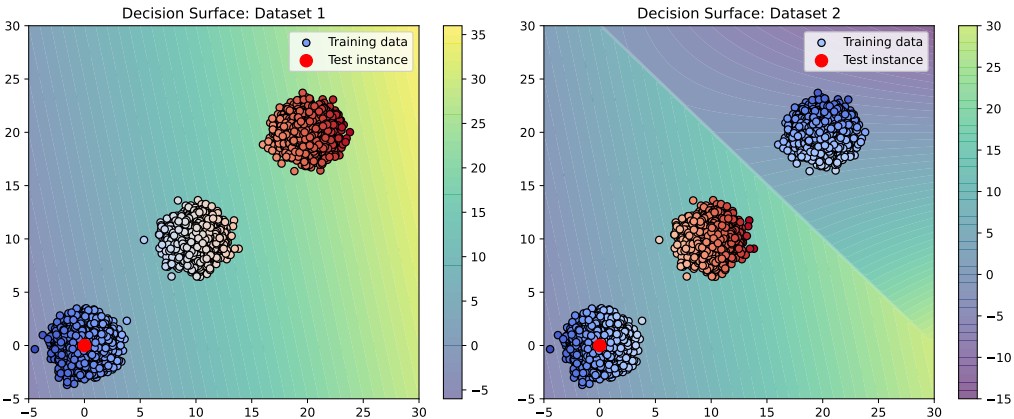

Figure 9: Decision surface visualization of GTMs trained on Dataset 1 (left) and Dataset 2 (right). The test instance at $[0, 0]$ is marked in red.

### A.8 DATASET LICENCES

All datasets used in this study (Wine Quality (Cortez et al., 2009), Digits (Pedregosa et al., 2011), and the Parkinson's Telemonitoring Tsanas & Little, 2009 dataset from UCI Machine Learning Repository) are licensed under a Creative Commons Attribution 4.0 International (CC BY 4.0) license.

## A.9    COMPUTE RESOURCES

The computations were carried out on a FormatServer THOR E221 (Supermicro) server equipped with two AMD EPYC 7702 64-Core processors and 512 GB of RAM with operating system Ubuntu 22.04.1 LTS. All experiments were run using only 4 cores and 16GB of RAM. The experiments from case study section took less than 1 CPU hour Model training was computationally efficient, requiring less than 4 CPU hours in total. However, the comprehensive ranking calculation across all metrics, models, and datasets was more resource-intensive, consuming approximately 1000 CPU hours. The finetuning of metrics hyperparameters took around 200 CPU hours.

