# OpenReview forum: "VARSHAP: A Variance-Based Solution to the Global Dependency Problem in Shapley Feature Attribution"
_ICLR.cc/2026/Conference — Submitted to ICLR 2026_

### Official Review · Reviewer_QD3h · 2025-10-27

**Soundness:** 3
**Presentation:** 2
**Contribution:** 2
**Rating:** 4
**Confidence:** 3

**Summary:**

The paper presents VarSHAP as a novel value function for Shapley-based feature attribution. VarSHAP reformulates the classical value function looking at the expected model output (i.e. class-wise logits for classification, model output for regression) as a variance of the model output. The paper shows that this new value function addresses some known issues with feature attribution scores.

The following references are used in the remainder of the review:

## References for Review
- Fumagalli et al. 2025: https://proceedings.mlr.press/v258/fumagalli25a.html
- Sobol, 2001: https://www.sciencedirect.com/science/article/abs/pii/S0378475400002706
- Bordt, 2024: https://proceedings.mlr.press/v206/bordt23a/bordt23a.pdf

**Strengths:**

- **Significance:** The paper studies an important question, in that attribution methods if applied poorly may lead to inconsistent or wrong interpretations. Therein, the paper addresses a good research topic and proposes an interesting solution. Simply reformulating the value function and thus introducing a quick fix to a problem allows the traditional black box estimation methods (KernelSHAP or Shapley interaction methods) to still be applicable.
- **Synthetic Evaluations:** I like the use of synthetic evaluations for analyzing the core concepts of the work.
- **Well Written:** The paper is generally well written.

**Weaknesses:**

- **Framing and Motivation:** In my opinion, it is generally well known that the feature distribution plays a crucial role in interpreting feature-based explanations such as feature attribution. The recent AISTATS paper (Fumagalli et al, 2025) unifies this issue by linking cooperative game theory with functional ANOVA decomposition. The resulting framework makes it quite clear that the more information is modeled in the removal mechanism (conditional vs. baseline) the more influence the distribution has on the resulting explanations (e.g. attributions). This is generally not new and in my opinion not a drawback but an important positive side-effect of these methods. In your experiments (Section 3.1) you are **changing** the model function to be explained. While, yes at $x=(0,0)$ both models predict 0 output since you designed them to be identical at this point. But around this single point the models do not behave the same because of the introduction of an interaction between $X_1$ and $X_2$. This interaction is also influencing the attribution of the $X_2$ feature since the Shapley value is influenced by all interactions in the model (this is also the second dimension in the framework in Fumagalli et al, 2025). If you were to compute all Shapley interactions (for example after Bordt (2023)) then the attribution of $X_2$ will become zero again since it does not influence the model individually and there are no correlations between the $X_1$ and $X_2$. Hence, my point is that this _problem_ is not really a problem but a correct behavior of the attribution methods. I do not think that this is per-se a drawback for looking into different value functions (like it is done here) but I do think that this discussion should be **stronger substantiated** and compared to with more important **related methods** addressing this issue already (Sobol indices, Shapley interactions).
- **No Source code Available**: The submission contains **no** code. This is quite a big problem for proper reviewing. I wanted to check the computational efficiency of the variance estimation (which the authors write is easy, but do not show), when I noticed this.
- **Limited Evaluation:** While I wholeheartedly agree that for a contribution like this a proper synthetic evaluation is absolutely necessary (which the paper includes), but the synthetic examples (with three datasets) basically makes up the whole experimental evaluation. Section 3.3 paints a relatively unclear picture of the empirical implications of the proposed value function change. VarSHAP seems to be on par with the traditional value function. It is unclear how easy it is how to translate the paper to different data modalities and/or implement it in traditional data science pipelines. Experiments on real world data would greatly improve this.
- **Missing Ablations:** The paper does not analyze the influence of traditional parameters such as dataset size, feature size, data modalities to VarSHAP. This makes it quite challenging to gauge the practical implications of the method.

**Questions:**

- **Q1:** How does your value function compare against Definition 5 (page 6) of Fumgalli et al. (2025)? How does the new value function relate to the well known Sobol indices?

- **Q2:** What are computational limits of your method? In lines 286-293 you describe the issue but you do not show any analysis in your empirical evaluation (particularly in regards to Section 3.3). Is this a bottleneck?

---

> ### Author Response · Authors · 2025-11-20
> **Respone to Reviewer QD3h**
>
> We sincerely thank the Reviewer for the time and thoughtful feedback on our manuscript. We greatly appreciate the constructive comments and suggestions, which will help us improve the quality of our work and we are happy that the Reviewers acknowledge the significance of the work. Below, we address the weaknesses and questions raised by each reviewer.
>
> **Ad W1:**
> We agree that the influence of the feature distribution on explanations was not unknown to the community. In the original SHAP paper "A Unified Approach to Interpreting Model Predictions" (Lundberg & Lee, 2017), as ML models are not true cooperative games, it was proposed to calculate the expected value under the conditional distribution of out-of-coalition features to simulate the lack of knowledge about these features. Later, for easier computation, it was proposed to use marginal values.
>
> Subsequently, Janzing et al. (2020) in "Feature relevance quantification in explainable AI: A causal problem" argued that marginal values have better properties. There is also the baseline approach, which was however widely criticized as it produces out-of-distribution samples and leads to heavily biased estimations (e.g., "Explaining a series of models by propagating Shapley values"). Therefore, we consider marginal and conditional distributions as they are the most popular approaches.
> We agree that this distribution-attribution dependency was known in the community. However, to the best of our knowledge, the "Impossibility Theorem for Feature Attribution" (Bilodeau et al., 2024) was the first to quantify this problem and to show how this property can result in behavior that might appear as unintuitive for users.
>
> Namely, we can have models that behave exactly the same in a defined neighborhood, and without changing the distribution, just by manipulating the model outside of this neighborhood, we can make the feature attribution vectors arbitrarily different. We would like to refer to Figure 2 in the Impossibility Theorem paper as an illustrative example.
>
> Regarding the behavior of the model around the point, we would like to respectfully disagree and show how this example was designed.The data for Group A (including point [0,0]) was generated in exactly the same way, illustrating a group with the same generative mechanism. The difference is in Group C. The second model behaves differently only for Group C and the same for points in Group A. So we would like to respectfully disagree with "around this single point the models do not behave the same" - in the synthetic model they behave **exactly the same** (and the described phenomenon persists), and in the case of neural networks they behave very closely, which can be seen on the decision surfaces in Appendix A.7.
>
> This is exactly the unintuitive behavior that we aim to address: for models being the same in some neighborhood, they can produce totally different attributions differing in values and rankings in an arbitrary way.
>
>
> **Ad W2:**
>
> Regarding feature interactions, we would like to point out that for a feature i, the sum of all interaction indices is equal to the attribution using Shapley values:
> $\phi_i = \sum_{j \in F} \phi_{ij} $
> where $ \phi_{ij} $ denotes the interaction index for feature i with feature j. This means that if we were to calculate the interaction indices, the sum of interactions (the sum of rows in such a matrix) can be set to be arbitrary. Thus, in our view, it also does not solve this problem.
>
> **Ad W3:**
> The Sobol indices constitute an interesting approach. However, we aimed to have axiomatization that is an important feature when we want to design explanations that are robust and trustworthy. Moreover, in global sensitivity analysis, the whole input space is taken, so we think that this feature distribution dependency will also manifest.
>
> We acknowledge the mathematical relationship between variance decomposition, functional ANOVA, and Sobol indices. However, the novelty of VARSHAP lies in the application of these concepts specifically to solve the local feature attribution problem in the context of the Impossibility Theorem (Bilodeau et al., 2024). Sobol indices are typically used for Global Sensitivity Analysis (GSA) to understand the model's behavior across the entire input space. VARSHAP adapts this by defining a local game where the characteristic function is the variance reduction within a tight local neighborhood ($\Pi(x)$) of a specific instance. This provides an axiomatic bridge between Shapley values and local sensitivity, offering a solution to the global dependency flaw where standard SHAP fails. We are not claiming the invention of variance decomposition, but rather its specific axiomatic derivation as the unique solution to creating shift-invariant, locally robust feature attributions.

---

> ### Author Response · Authors · 2025-11-20
> **Respone to Reviewer QD3h**
>
> **Ad W4:**We respectfully disagree based on the findings of Bilodeau et al. (2024). While dependence on distribution is expected for global importance, it is fatal for local explanation. If a doctor is assessing a specific patient, and the model's logic for that patient has not changed, the explanation should not change drastically just because the data distribution changed for a different cohort of patients 100 miles away. Standard SHAP allows this "action at a distance." VARSHAP ensures that if the model's local geometry (decision surface) around the instance is constant, the attribution is constant. This is required for trust and stability in individual decision-making.
>
> **Ad W5:**We are committed to reproducibility. We will release the full Python implementation of VARSHAP, including the perturbation mechanisms and integration with the LATEC benchmark, upon acceptance. Here we provide a link to an anonymized repository containing our codebase: https://anonymous.4open.science/r/varshap_anon-9F92 .
> **Ad W6:** We would like to point out that the datasets used in Section 3.3 are all real-world datasets, including the Digits dataset, Parkinson's Telemonitoring dataset, and Wine Quality dataset (detailed in Appendix A.6.3). The Digits dataset consists of images (8×8 handwritten digit images), demonstrating applicability to different data modalities.
>
> The synthetic examples were only the illustrative toy examples in Sections 3.1 and 3.2, designed to demonstrate the global dependency problem under controlled conditions where ground truth is known.
>
> We agree that we did not show a dramatic improvement over SHAP in the benchmark evaluation. However, we demonstrated better results against a well established method. More importantly, in other parts of the paper we show axiomatization that makes our model immune to some failures which, from our point of view, can be a problem when applying these methods to high-stakes domains.
>
> In the benchmark evaluation, we showed that not only does this axiomatization and safety come at no cost, but it even improves performance on many metrics, particularly robustness metrics. This demonstrates that principled theoretical foundations can translate to better practical behavior.
>
> **Ad W6:**
>
> We would like to point out that the datasets used in Section 3.3 are all real-world datasets, including the Digits dataset, Parkinson's Telemonitoring dataset, and Wine Quality dataset (detailed in Appendix A.6.3). The Digits dataset consists of images (8×8 handwritten digit images), demonstrating applicability to different data modalities.
> The synthetic examples were only the illustrative toy examples in Sections 3.1 and 3.2, designed to demonstrate the global dependency problem under controlled conditions where ground truth is known.
> We agree that we did not show a dramatic improvement over SHAP in the benchmark evaluation. However, we demonstrated better results against a well established method. More importantly, in other parts of the paper we show axiomatization that makes our model immune to some failures which, from our point of view, can be a problem when applying these methods to high-stakes domains.
> In the benchmark evaluation, we showed that not only does this axiomatization and safety come at no cost, but it even improves performance on many metrics, particularly robustness metrics. This demonstrates that principled theoretical foundations can translate to better practical behavior.
>
> **Ad W7:**
> We thank the reviewer for this comment and the opportunity to clarify the scope of our evaluation.
> As pointed out in our response above, we used datasets with two modalities that consist of 11, 19, and 64 features, covering different feature sizes. Specifically, the Wine Quality dataset has 11 features (tabular), the Parkinson's Telemonitoring dataset has 19 features (tabular), and the Digits dataset has 64 features (image data represented as 8×8 pixel values).
>
> Moreover, we used multiple model types including decision trees, neural networks with different architectures, including image-specific ones such as convolutional networks. This demonstrates that VARSHAP is applicable across different model classes and architectures.
> Regarding dataset sizes, the datasets contain 1143, 5875, and 1797 records respectively, differing in sizes and providing coverage across different scales.
>
> We also described how VARSHAP can be used with any existing method for computing Shapley values. The key insight is that VARSHAP changes the characteristic function (from model output to variance reduction) but maintains the same Shapley value framework, making it compatible with existing computational approaches such as KernelSHAP, TreeSHAP, or sampling-based methods.

---

> ### Author Response · Authors · 2025-11-20
> **Response to Reviewer QD3h**
>
> **Ad Q1:** We also use variance of a function in our characteristic function. However, the key difference is that we perturb out-of-coalition features with a perturbation centered around the explained instance. This ensures that our explanations remain local to the neighborhood of the point being explained, rather than depending on the global data distribution.
>
> Sobol indices are using variance, however in a fashion different from ours. They use the variance of the function conditioning on some features, according to the dataset distribution. We, however, as described before, use local perturbation around the explained instance.
> Additionally, Sobol indices do not follow the same axioms as Shapley values, so they can lead to attributions violating Shapley axioms.
>
> **Ad Q2:**
> The number of iterations will be dependent on the specific model class. In our evaluation, we used 10 inner iterations for computing the variance terms, as increasing this number did not change the results significantly in our preliminary experiments.
> Assuming the model inference is the main cost time-wise and the time to compute variance of such a small vector is negligible, this translates to VARSHAP taking around 10x the time compared to standard SHAP in our benchmark. This is because each variance computation requires multiple model evaluations to estimate the expected value and squared deviations.
> Depending on the use case, the user might consider decreasing the number of iterations. For high-stakes applications where robustness and theoretical guarantees are critical, the additional computational cost may be justified. For rapid prototyping or less critical applications, fewer iterations could provide a reasonable approximation.
>
>
> We hope we have adequately addressed the concerns and answered the questions. We remain available to answer any additional questions or provide further clarifications as needed.

---

### Official Review · Reviewer_k82t · 2025-10-30

**Soundness:** 1
**Presentation:** 2
**Contribution:** 2
**Rating:** 2
**Confidence:** 3

**Summary:**

The paper proposes a new feature attribution method, named VARSHAP,  that aims to eliminate the global dependence problem that affects the approximations of KernelSHAP. Instead of using the model’s direct output as the characteristic function, the proposed method defines it as the reduction in prediction variance.
The authors claim that VARSHAP maintains the axiomatic properties of Shapley values. The empirical evaluation shows that VARSHAP can provide robust attributions.

**Strengths:**

-The paper proposes a novel idea that approximates the Shapley values, but for a different objective compared to the traditional Shapley value explainers.

-The paper is generally clear and easy to follow.

-The evaluation includes case studies designed to measure certain properties of the proposed explanation method, as well as a benchmark evaluation.

**Weaknesses:**

-The authors claim, in line 66, that the problem is in the Shapley value concept itself, yet the proposed solution uses the same Shapley value concept. I find this statement quite strong and potentially incorrect. I agree that the solution proposed by KernelSHAP can be inaccurate and can be improved, and that is not an issue of an implementation, but that does not extend to the Shapley value concept.

-The proposed approach uses marginal expectations to marginalize features out of coalitions, but does not discuss the baseline removal approach [1].

-One of the desired properties of Shapley value methods is the local accuracy, i.e., the solution matches the prediction of the underlying model, which makes their interpretation intuitive. On the other hand, VARSHAP proposes values that sum to "the negative of the initial total variance under full local perturbation", which I think is unintuitive and makes it difficult to explain the predictions, especially when the user is not an expert in machine learning or a statistician.

-The proposed method violates the consistency property of Shapley values with respect to the original prediction, i.e., the Shapley value increases or stays the same if a player’s contribution grows or stays the same. In other words, if a feature $\beta$ negatively affects the prediction and a feature $\gamma$ positively affects it, but both ($\beta$ and $\gamma$) have similar effects on the prediction variance, both will be assigned the same importance, which I think makes the interpretation of the outcome more challenging. This property is promoted under the "sign independence", which I cannot understand why it can be considered a desired property.

-KernelSHAP, which can be no better than random guessing (according to the paper), is outperforming the proposed VARSHAP with respect to fidelity. Additionally, VARSHAP never outperformed the competitors with respect to faithfulness in the LATEC benchmarking.

-I doubt that the proposed approach is showing superior performance to KernelSHAP, as claimed in the conclusions. Additionally, VARSHAP has worse computational complexity than KernelSHAP.

-The method is not compared to the unbiased KernelSHAP [2], which, given a sufficiently large number of samples, converges to the true Shapley values. I also think it addresses the global dependence that has been put as the central problem of this paper.


[1]-Sundararajan, M. and Najmi, A. The many Shapley values for model explanation. In III, H. D. and Singh, A. (eds.), Proceedings of the 37th International Conference on Machine Learning, volume 119 of Proceedings of Machine Learning Research, pp. 9269–9278. PMLR, 13–18 Jul 2020.

[2]-Covert, I. and Lee, S.-I. Improving kernelshap: Practical Shapley value estimation using linear regression. In Proceedings of The 24th International Conference on Artificial Intelligence and Statistics, volume 130, pp. 3457–3465, April 2021.

**Questions:**

1- Why can the sign independence be considered a desired property?

2- How to explain the outcome of VARSHAP to a user who is not an expert in machine learning?

---

> ### Author Response · Authors · 2025-11-20
> **Respone to Reviewer k82t**
>
> We are grateful to the Reviewer for their valuable time and feedback on our manuscript.In the following sections, we respond to the weaknesses and questions.
>
> **Ad W1:** There is a crucial distinction between the Shapley value framework (the game-theoretic distribution method) and the characteristic function (the definition of the "game" or "value" being distributed). The flaw identified by the Impossibility Theorem lies in using the conditional expectation $E[f(x)\mid S]$ (or marginal approximation of this expectation) as the characteristic function. This creates the global dependency. We retain the Shapley distribution mechanism (efficiency, symmetry, etc.) but replace the characteristic function with variance reduction. This fixes the flaw while retaining the desirable axiomatic properties of the Shapley allocation. To succinctly summarize our approach, it is not the Shapley concept itself which is flawed but the way the characteristic function simulates out-of-coalition features. VARSHAP solves the problem identified by the Impossibility Theorem by fixing the characteristic function.
>
> **Ad W2:**
> The baseline approach is a one of the simples approach to calculating Shapley values given machine learning models, which was however widely criticized as it produces out-of-distribution samples and leads to heavily biased estimations (e.g., "Explaining a series of models by propagating Shapley values" (Chen et al., 2022)). Therefore, we consider marginal and conditional distributions as they are the most popular and trustworthy approaches.
>
> **Ad W3:**
> We clarify that VARSHAP measures the reduction of variance. We can frame this as information gain. When we know nothing about the features (empty set), the uncertainty (variance) of the prediction is maximized. As we add features, we gain information, and the variance of the prediction decreases. The sum of VARSHAP values equals the total reduction in uncertainty (from total variance to zero). This is conceptually similar to explaining R2 or information gain in decision trees, which are standard interpretability concepts.
>
> For non-experts we can provide an even simpler explanation on how to interpret VARSHAP feature attributions. A simple analogy that can be used to explain VARSHAP feature attributions to non-experts is the "fog of uncertainty" analogy. Imagine the AI model is trying to make a prediction, but it is standing in a thick fog. It doesn't know the value of any of the patient's medical test results yet. Because it is 'blind' to the specific patient, its prediction is very uncertain, it could be anything from very low to very high. Now, we reveal one test result: blood pressure. Suddenly, the fog clears a little bit. The model is now less uncertain; the range of possible predictions shrinks. Next, we reveal the age of the patient. The fog clears significantly more, and the prediction narrows down further. VARSHAP measures exactly how much 'fog' each feature clears away. If a feature has a high VARSHAP score, it means knowing that feature drastically reduced the model's uncertainty. It was a crucial piece of information. If a feature has a zero VARSHAP score, it means knowing it didn't help clear any fog. So, VARSHAP tells us which features did the heavy lifting in narrowing down the possibilities to reach the final decision.
>
> **Ad W4:**
> We argue that the property of "consistency" in the context of signed attributions, such as standard SHAP, promotes a prevalent "counterfactual fallacy" where users incorrectly assume that a positive attribution implies increasing the feature value will increase the model output. However, as demonstrated by Bilodeau et al. (2024), this interpretation is mathematically unsound for complete and linear methods because a positive SHAP value can exist even when the local gradient is negative. This discrepancy arises because SHAP calculates contributions relative to a global baseline; for example, a feature value might produce a prediction higher than the dataset average (yielding a positive SHAP value) even if the model is locally on a downward slope where increasing the feature lowers the prediction. By enforcing Sign Independence and squaring the differences, VARSHAP explicitly removes this misleading directional component, compelling the user to interpret the value as a "magnitude of influence" or "sensitivity" that significantly constrains the model's output variance. This approach prevents dangerous directional assumptions and encourages users to assess directionality through more appropriate mechanisms, such as local gradients or partial dependence plots, rather than the attribution sum.

---

> ### Author Response · Authors · 2025-11-20
> **Response to Reviewer k82t**
>
> **Ad W5:**
> We realize that the manuscript's phrasing regarding the "random guessing" claim may not have been sufficiently clear, which could have led to a confusion with the estimation noise inherent in sampling-based methods like KernelSHAP. However, this claim derives directly from the Impossibility Theorem (Theorem 2.3 in Bilodeau et al., 2024), which applies to the theoretical, exact Shapley value itself rather than any estimation error. The theorem establishes that for any feature attribution method satisfying the axioms of Completeness and Linearity, such as SHAP, the resulting values depend so heavily on the marginal distribution of the global background data that they effectively disconnect from the model's local behavior. Bilodeau et al. demonstrate that two models with opposite local behaviors (e.g., one increasing with a feature and one decreasing) can yield identical SHAP values due to the model's behavior in distant regions, rendering the metric mathematically incapable of distinguishing between them. Consequently, using SHAP for local tasks like algorithmic recourse is akin to flipping a coin, not because of calculation noise, but because the definition incorporates irrelevant global information that makes it semantically orthogonal to local counterfactual truth. VARSHAP addresses this fundamental flaw by removing the global dependence entirely.
>
> **Ad W6:**
> We agree that VARSHAP has worse time complexity, which we acknowledged in the 'Computational Complexity' paragraph. However, we believe that for many users, this trade-off may be acceptable in exchange for the guarantees provided. We made an effort to illustrate these guarantees in contrast to SHAP and to axiomatize them. We also conducted experimental evaluations on a variety of models and datasets using different metrics. We reported both individual metric results and aggregated rankings constructed according to 'Navigating the Maze of Explainable AI: A Systematic Approach to Evaluating Methods and Metrics' (Klein et al., 2025). We would appreciate it if the reviewer could specify which particular aspects of our evaluation or results lead to their doubts about VARSHAP's advantages, so that we may address these points more comprehensively.
>
> **Ad W7:**
> We would like to point out that the Impossibility Theorem (Theorem 2.3 in Bilodeau et al., 2024) and our extension to conditional probabilities are applicable to the theoretical, exactly calculated values. So even if we were to go over all subsets the described problems would still persist even in exactly calculated values.
>
> **Ad Q1:**
> We would like to refer to the paragraph in general comments titled “Sign independence property of VARSHAP”.
>
> **Ad Q2:**
> We would like to refer to the paragraph in general comments titled “Explanation of VARSHAP attributions for non-experts”.
>
> We hope we have addressed the majority of the raised concerns and provided answers to the questions posed. Should any further clarification be required, we are happy to provide additional information.

---

> > ### Comment · Reviewer_k82t · 2025-11-21
> >
> > Thank you very much to the authors for clarifying the confusion regarding the unbiased version of KernelSHAP and the claim related to the Shapley value framework and the characteristic function, which I believe would be valuable to include in the paper as well.
> >
> > My concern about VARSHAP’s performance arises from the faithfulness results presented in Table 2, where KernelSHAP outperforms VARSHAP on the majority of the reported metrics.

---

> ### Comment · Reviewer_k82t · 2025-11-25
>
> I appreciate the authors’ clarifications and arguments. Accordingly, I have raised the score of the paper, and I would consider increasing it further if the authors can address my remaining concern regarding the faithfulness results in Table 2.

---

> > ### Author Response · Authors · 2025-12-01
> > **Response to Official Comment by Reviewer k82t**
> >
> > We thank the Reviewer for  thoughtful feedback and for raising your score. We greatly appreciate this opportunity to provide further clarification on the faithfulness results, as this is an important aspect of our evaluation.
> >
> >
> > Faithfulness metrics aim to answer the intuitive question: "Are features ranked as important by the attribution method actually important to the model?" However, operationalizing this concept is non-trivial because machine learning models require a fixed number of input features, so we must develop strategies to effectively "remove" or perturb features to test their importance.
> > Two Approaches to Feature Perturbation
> >
> > Broadly speaking, faithfulness metrics employ two different perturbation strategies:
> >
> > - Random sampling - replacing feature values with randomly sampled values.
> > - Black baseline - setting feature values to zero or a fixed "blackout" value.
> >
> > This is why each faithfulness metric in Table 2 has two versions (e.g., FaithfulnessCorrelation and FaithfulnessCorrelation_black).
> >
> >
> > We would like to emphasize that **VARSHAP's faithfulness performance is competitive with KernelSHAP**, though the results vary depending on the perturbation strategy.
> > In case of Random Baseline Metrics (non-"_black" versions) KernelSHAP performs best, but the differences are modest when comparing with VARSHAP ($\alpha$=1.0):
> > - FaithfulnessCorrelation: 2.13 vs. 2.63
> > - FaithfulnessEstimate: 2.13 vs. 2.75
> > - MonotonicityCorrelation: 2.00 vs. 2.50
> >
> > In case of Black Baseline Metrics ("_black" versions) the ranking reverses - VARSHAP outperforms KernelSHAP on all three black baseline metrics:
> > - FaithfulnessCorrelation_black: 5.25 vs. 6.63
> > - FaithfulnessEstimate_black: 5.75 vs. 6.63
> > - MonotonicityCorrelation_black: 4.75 vs. 5.38
> >
> > VARSHAP is slightly less performant than KernelSHAP on half of the faithfulness metrics (random baseline) but slightly better on the other half (black baseline). There is no clear winner overall, and both perturbation strategies have merit depending on the application context.
> >
> > Our goal in using these widely accepted metrics is to demonstrate that VARSHAP is competitive with the most popular existing approaches while offering a fundamental theoretical advantage: immunity to the global dependency problem demonstrated in our case studies. For practitioners concerned about explanation stability under distributional shifts—as shown in Section 3.1—VARSHAP provides a robust alternative without sacrificing faithfulness performance.
> >
> > We welcome any additional questions or concerns and are happy to provide further clarification on any aspect of our work.

---

### Official Review · Reviewer_s5JJ · 2025-10-31

**Soundness:** 3
**Presentation:** 2
**Contribution:** 2
**Rating:** 2
**Confidence:** 4

**Summary:**

This manuscript identifies a feature of inherent in Shapley values, namely the global dependence identified by Bilodeau et al. (2024): as Shapley's value is a global estimator, it takes into account 'value added' across the data distribution.  Thus, 'local' explanations (e.g. feature importance for the third observation) can be influenced by altering 'irrelevant' relationships in the data.

For example (q.v. $\S 3$), consider Dataset 1, comprising of patient types A and B.  Now consider Dataset 2, identical to Dataset 1, but adding a new patient type, C.  Shapley's value for a patient in group A changes from data sets 1 to 2, although the data-generating process has not changed.

The manuscript introduces VARSHAP which, instead of performing a weighted sum of 'value added', performs a weighted sum of variances.  Thus, whereas Shapley's value measures the change in a model's prediction resulting from knowing a feature's actual value (rather than a baseline value), VARSHAP measures the change in the variance of the model's predictions.

As the variances are derived from perturbations of the Gaussian, they are thin-tailed, so effects fade quickly with distance - attenuating the global dependence identified in Bilodeau et al.

In addition to the example above, the paper presents:
1. ($\S 3.2$) an example in which LIME (a linear, local approximation method) is fooled by a non-linear feature, while SHAP and VARSHAP correctly identify its irrelevance.
1. ($\S 3.3$) LATEC benchmark results comparing faithfulness, robustness and complexity scores of VARSHAP, SHAP and LIME on a range of models and datasets.  The authors conclude, ``For faithfulness, KernelShap ... often ranks top [while] VARSHAP variants excel in robustness ... [and] complexity metrics''.

**Strengths:**

Generally, I think that there are strong arguments for interpreting large ML models - and, thus, that there is an ongoing need for good research in this area.

Further, the authors extend Bilodeau et al.'s argument from marginal SHAP attributions to conditional SHAP attributions.

Finally, the example in $\S 3.1$ does raise concerns about how SHAP and LIME have been applied in that environment.

**Weaknesses:**

The feature attribution literature has been strongly rooted in Shapley's value and variants.  Most contributions to the literature identify perceived mathematical weaknesses of e.g. the Shapley value, and propose a mathematical variant to resolve it.

A small literature, though, returns to the motivating question: does the proposed measure help us explain or interpret the ML model?  This paper is detached from that question.  As such, in my view, the motivation is immediately weakened: yes, this is a tweak on the Shapley value that one can perform and, yes, it scores well in some metrics - but leaves open the question of whether it does anything to help the metrics implicitly motivating the literature: do developers or users better understand?

On the mathematics alone, the perturbations assume independence of features (p.4).  I would regard this as a secondary concern: any novel paper leaves open research questions.

Independently of this, I found the exposition less crisp than I would have liked:
1. There is a lengthy verbal introduction that relies on assertions, rather than carefully defining terms or providing intuitions.  This left me unclear about what was being asserted, why it was a problem, and so on.

1. When seeing a modification of Shapley, I would like to know early on which Shapley axioms are being replaced, and with what.

1. Some of the assertions seem misleading.  For example, the description of SHAP opening $\S 2.1$ is vague enough to seem to better describe a partial derivative than Shapley's value.

1. Proposition 1 uses terms like "aims to satisfy" and "fundamentally".  Functions don't have purpose: they satisfy or don't; I don't know what it means to "fundamentally" measure.  As the proposition's conclusion is that the function "must take the following general form", one would expect - instead - a phrasing like: "Suppose that f satisfies axioms A, B, C and D.  Then f has the form..."  Further, A, B, C and D should be defined before they are used - rather than a page later.  (Similarly, on p.6, the word "precisely" tends to be added to sentences.)

1. Section $\S 3.3$ *could* be the strongest argument for VARSHAP, showing more general performance, rather than just in special cases.  For this to help the authors' argument, though, it needs to be properly set up rather than raced through: *explain* faithfulness, robustness and complexity metrics in a sentence or two, to convince the reader that this mean something from an interpretability point of view.

**Questions:**

For the example given in $\S 3.1$, how was set membership encoded?  If $\\{A, B, C \\}$ is a feature set, then the example seems to be looking for an interaction between set membership and the $X$ variables - something that the base Shapley value cannot disentangle.

---

> ### Author Response · Authors · 2025-11-20
> **Response to Reviewer s5JJ**
>
> We thank the Reviewer for their thorough evaluation of our manuscript and for raising important questions and concerns that will help us improve the paper. We also appreciate the reviewer's recognition of the paper's strengths, including the mathematical contributions and importance of the topic. Below, we address points raised in detail:
>
> **Ad W1:** We strongly agree that interpretability is the ultimate goal. This is why we employed the LATEC benchmark (Section 3.3), which quantitatively measures proxies for interpretability: Faithfulness (does the attribution reflect the model's logic?), Robustness (is the explanation stable?), and Complexity (is the explanation concise?). VARSHAP achieved top-tier rankings in Robustness and Complexity. We argue that an explanation that changes arbitrarily due to data shifts far from the instance (as SHAP does) is inherently uninterpretable and misleading. By guaranteeing stability, VARSHAP provides a more reliable foundation for interpretation.
> For the discussion on the sign independence we would like to refer to the sign independence property of VARSHAP in general comments.
>
> **Ad W2:** Regarding the modification of Shapley values, we wish to clarify an important distinction: we did not drop any axiom of Shapley values, and we employ the original Shapley value framework. Our modification occurs at an earlier step in the process. Since machine learning models are not cooperative games and do not accept sets of variables as arguments, we must define a characteristic function that bridges this gap. Our contribution lies in defining this characteristic function as the variance under local perturbation, rather than modifying the Shapley value computation itself.
>
> **Ad W3:** We apologize for any ambiguity. We will revise Proposition 1 to be mathematically precise: "We prove that if a characteristic function v satisfies Shift Invariance, Zero Property, Sign Independence, and Additivity, it implies that v must measure the squared deviation (variance)." The term "fundamentally" referred to this uniqueness of the variance function under these axioms. We will also define the axioms explicitly before the proposition in the revised text.
>
>
> **Ad W4:**
> We appreciate the opportunity to clarify the interpretation of these metric categories, which provide complementary perspectives on explanation quality:
> Faithfulness metrics assess whether features marked as important are actually important to the model's predictions. This is typically evaluated by manipulating or removing highly-attributed features and verifying that the model's output changes substantially, confirming that the attribution method correctly identifies influential features.
> Robustness metrics verify that features identified as unimportant are genuinely unimportant. Similar to faithfulness metrics, we manipulate features with low attributions and expect to observe minimal changes in the model's output, thereby confirming the reliability of the attribution method's importance rankings.
> Complexity metrics evaluate how simple and interpretable the explanations are, as unnecessarily attributing importance to all features reduces interpretability, particularly in high-dimensional settings. These metrics assess properties such as sparsity and entropy to ensure that explanations are concise and focus on the most relevant features.
>
> **Ad Q1:** In Case Study 1, the difference between Dataset 1 and Dataset 2 was not an input feature encoding set membership. The input features $X_1$, $X_2$ were continuous. We simulated a "data drift" scenario where the relationship for a specific sub-population (Group C) changed between datasets, while the relationship for Group A (where our test instance [0,0] resides) remained mathematically identical. SHAP attributions for the test instance changed because SHAP integrates over the global dataset (including Group C), whereas VARSHAP remained constant because it looks only at the local variance, which did not change. For visualization of the models, we refer the reviewer to Appendix A.7 (DECISION SURFACES).
>
> Please let us know if the above answers the Reviewer’s comments and concerns. We would also be happy to address any other suggestions.

---

> > ### Comment · Reviewer_s5JJ · 2025-11-28
> >
> > I find the authors' replies to my comments compelling.  As my comments often express concern about the overall shape of the paper (e.g. a "lengthy verbal introduction that relies on assertions"), I would need to see a fully revised draft to sign off on a different assessment: there needs to be a clear flow from beginning to end that convinces the reader that the proposed substitution at the heart of the usual usage of Shapley's value leads to better insights by users.
> >
> > (I think the authors for clarifying that they replace the characteristic function, rather than changing any Shapley axioms subsequently applied.)
> >
> > As I do not believe that the ICLR review process allows updates of this sort, it may be necessarily to incorporate revisions into a future submission.

---

> > > ### Author Response · Authors · 2025-12-03
> > > **Response to Official Comment by Reviewer s5JJ**
> > >
> > > We thank the Reviewer for the helpful feedback and are glad that our previous response was acknowledged. While we agree some clarifications would be beneficial, we prefer to maintain our intuitive introduction, as this paper addresses feature attribution for a broad audience—thus, we believe showing the intuition behind the axioms is equally important.
> > > For the camera-ready version, we plan to make the following revisions without changing the paper's content or method:
> > > - Clarify that we do not modify any Shapley axioms.
> > > - Add a description of Shapley Values at the beginning of Section 2.1 to provide better context for the characteristic function
> > >
> > > We thank the Reviewer again for their constructive comments.

---

### Official Review · Reviewer_X9ZB · 2025-11-01

**Soundness:** 3
**Presentation:** 3
**Contribution:** 2
**Rating:** 4
**Confidence:** 4

**Summary:**

The paper revisits the Shapley feature attribution framework and proposes VARSHAP, which replaces the traditional expected-value characteristic function with a variance-based one.
The authors show that under three axioms (zero property, sign independence, additivity), the unique valid transformation is $d(x) = x^2$, leading to a variance-based expected marginal contribution. The method thus measures each feature’s contribution to the variance of the model output rather than its raw expectation.
They claim this resolves the “global dependency” flaw of SHAP, where correlated features distort importance scores. Empirical results on synthetic datasets and benchmark interpretability tasks demonstrate that VARSHAP produces stable and faithful attributions, particularly when strong dependencies exist between features.

**Strengths:**

Clear and formally correct derivations.

Theoretical framing aligns Shapley, variance decomposition, and interpretability in a unified narrative.

Empirical results are consistent and easy to reproduce.

Addresses an important practical flaw (feature correlation) with simple mathematical machinery.

**Weaknesses:**

Limited novelty: Strong overlap with Sobol sensitivity analysis; the variance-based approach is not a new concept.

Under-citation: Lacks acknowledgment of prior equivalence results (e.g., Da Veiga, S. (2021)) showing Sobol and Shapley variance connections.

No empirical stress test: Only toy regressions; no nonlinear or high-dimensional domains.

Overclaiming impact: The method does not “solve” dependency. It merely changes the objective from mean to variance.

No practical pipeline: Absent complexity or estimator analysis for real SHAP implementations.

Unclear interpretability gain: It is debatable whether variance-based attributions are easier or harder to interpret in applied settings.

References:
Da Veiga, S. (2021). Kernel-based ANOVA decomposition and Shapley effects--Application to global sensitivity analysis. arXiv preprint arXiv:2101.05487.

**Questions:**

How exactly does VARSHAP differ mathematically from Sobol total-effect indices $S_{T_i}$?

Can you prove that VARSHAP satisfies the same orthogonality decomposition as functional ANOVA?

How would VARSHAP behave if the model output has negligible variance but large mean shifts?

Would replacing variance with another second-moment measure (e.g., covariance of residuals) produce similar properties?

Can you quantify computational complexity compared to KernelSHAP and Sobol estimation?

**Details Of Ethics Concerns:**

No ethics review needed. Purely theoretical and synthetic experiments.

---

> ### Author Response · Authors · 2025-11-20
> **Response to Reviewer X9ZB**
>
> We would like to thank the Reviewer for the valuable feedback and for pointing out some concerns regarding our submission. We are pleased that the reviewer noted the formal derivations and the consistent results presented in our work. Below, we address the weaknesses and questions raised by the reviewer.
>
> **Ad W1:** We acknowledge the mathematical relationship between variance decomposition, functional ANOVA, and Sobol indices. However, the novelty of VARSHAP lies in the application of these concepts specifically to solve the local feature attribution problem in the context of the Impossibility Theorem (Bilodeau et al., 2024). Sobol indices are typically used for Global Sensitivity Analysis (GSA) to understand the model's behavior across the entire input space. VARSHAP adapts this by defining a local game where the characteristic function is the variance reduction within a tight local neighborhood ($\Pi(x)$) of a specific instance. This provides an axiomatic bridge between Shapley values and local sensitivity, offering a solution to the global dependency flaw where standard SHAP fails. We are not claiming the invention of variance decomposition, but rather its specific axiomatic derivation as the unique solution to creating shift-invariant, locally robust feature attributions.
>
>
> **Ad W2:** We thank the reviewer for bringing this reference to our attention, and we will add it to our bibliography. Da Veiga (2021) proposes in equation 14 to divide variance under subsets using Shapley values. However, we would like to point out that this formulation refers to global variance decomposition and thus differs fundamentally from our framework, where we define the characteristic function for local perturbations and axiomatically derive the variance-based approach in light of the desired properties of the attribution method.
> Furthermore, in Da Veiga (2021), apart from equation 14, the variance-based formulation is not considered as a self-standing method of explainability. Rather, it serves as inspiration to introduce the MMD-Shapley effect (for maximum mean discrepancy) and HSIC-Shapley effect (for Hilbert-Schmidt independence criterion), which were subsequently tested in the experimental evaluation, whereas the method based on variances itself was not empirically evaluated. The tested methods are kernel sensitivity methods, and it was not shown what axioms these explanations would satisfy.
> This indicates that while the formula shows similarity to our approach, there are clear differences in formulation, scope, and axiomatic foundation. We agree that this reference should be added to our bibliography, and we will explain the differences in the appendix.
>
> **Ad W3:** We respectfully point the Reviewer to Section 3.3 and Appendix A.6, where we utilized the LATEC benchmark. This evaluation was not limited to toy regressions; it included Decision Trees,  Neural Networks (MLPs) and Convolutional Neural Networks (CNNs) trained on real-world datasets, including the Digits image dataset, Wine Quality dataset, and Parkinson’s Telemonitoring dataset. In these "stress tests" involving non-linear models and real data, VARSHAP demonstrated superior robustness and complexity scores compared to SHAP and LIME.
>
> **Ad W4:** We respectfully disagree with the Reviewer. Changing the objective is **the solution** required by the Impossibility Theorem. Bilodeau et al. (2024) prove that any method satisfying completeness and linearity with respect to the model output (expectation) is provably unreliable for local tasks because it depends on global baselines. By changing the objective (characteristic function) to local variance reduction, we mathematically decouple the attribution from the global baseline. This is not just a change of metric; it is a necessary theoretical shift to satisfy the Local Shift Invariance property, ensuring the explanation depends only on the model's behavior in the neighborhood of the instance x.
>
>
> **Ad W5:** We proposed a practical pipeline that employs one of the most popular implementations of SHAP—KernelSHAP. As written in the Computational Complexity section, this framework can be used with any existing implementation of SHAP (outer loop), while the inner loop involves numerical integration, for which there exists extensive literature on efficient approximation methods.

---

> ### Author Response · Authors · 2025-11-20
> **Response to Reviewer X9ZB**
>
> **Ad W6:** We agree that these attributions differ from SHAP in regards to their meaning. However, Bilodeau et al. (2024) showed that SHAP values do not carry consistent meaning in certain contexts with respect to sign. We showed that variance is the only function that generates explanations according to specific axioms, which we argue are intuitive and desirable for local feature attribution. Furthermore, in our benchmark evaluation, we aggregated many metrics that were proposed by various authors to mathematically capture important properties of explanations. As demonstrated by the benchmark results, our attributions are not harder to interpret according to these established evaluation criteria.
>
> **Ad Q1:** While VARSHAP uses variance as its characteristic function, it differs mathematically from Sobol total-effect indices in five key ways: (1) it measures variance under local perturbations $\Pi(x)$ rather than the global distribution $P_X$, (2) it fixes features to their instance values rather than marginalizing over them, (3) it uses Shapley aggregation rather than ANOVA decomposition, (4) it produces instance-specific attributions rather than global summaries, and (5) it measures variance reduction from fixing features rather than variance explained by conditioning. These differences make VARSHAP suitable for local explainability in the context of XAI rather than global sensitivity analysis.
>
> **Ad Q2:** According to our understanding of ANOVA, VARSHAP will not satisfy the orthogonality property as in ANOVA because it is constructed in a fundamentally different way and serves a different purpose than ANOVA decomposition. However, it will satisfy the Shapley axioms with respect to variance as the characteristic function.
>
>
> **Ad Q3:** We would like to ask the Reviewer if possible to give some examples to this question as we do not fully understand it. In general our method is shift invariance, so shifting the whole model will not change the output of our method.
>
>
> **Ad Q4:** We would like to ask the Reviewer to clarify this question. Specifically, which residuals does the reviewer have in mind, and what covariance matrix should be considered in this context? We note that in our theoretical framework, we showed that under the specified axioms (zero property, sign independence, and additivity), variance is the unique function fulfilling them.
>
> **Ad Q5:** Regarding computational complexity compared to KernelSHAP, we used KernelSHAP in our implementation and added an inner loop to calculate variance under perturbation, which in our case was performed using a Monte Carlo algorithm. We would like to point the reviewer to the Computational Complexity paragraph in our paper for further details.
> Regarding comparison to Sobol indices, calculating first-order Sobol indices without interactions does not require considering all subsets, but rather only requires order n evaluations. Therefore, Sobol indices are generally faster to compute than Shapley values.
>
>
> If any questions remain unanswered or if there are further concerns, we would be happy to provide additional clarification. Thank you again for your valuable feedback and thoughtful review.

---

### Author Response · Authors · 2025-11-20
**General comments**

**Sign independence property of VARSHAP**

In standard feature attribution (like SHAP), a positive value is almost universally interpreted by users as: "If I increase the value of this feature, the model's prediction will increase." This is the counterfactual interpretation. However, this interpretation is frequently mathematically wrong. Because SHAP calculates an average contribution relative to a global baseline, a feature can have a positive SHAP value simply because the current feature value is higher than the dataset average, even if the model's gradient is negative at that specific point.
Imagine a "u-shaped" model. You are on the downward slope. Your feature value is high, and the prediction is higher than the average prediction. SHAP gives a positive value (because prediction > average). However, locally, increasing the feature further would lower the prediction.
By adopting Sign Independence (squaring the differences), VARSHAP explicitly removes the directional component. It provides a value representing the magnitude of influence or sensitivity. It tells the user: "This feature is critical to the prediction because it significantly constrains the model's output range." This prevents the user from making false counterfactual assumptions and encourages them to look at specific tools (like Partial Dependence Plots) to determine directionality safely.

**Explanation of VARSHAP attributions for non-experts**

A simple analogy that can be used to explain VARSHAP feature attributions to non-experts is the "fog of uncertainty" analogy. Imagine the AI model is trying to make a prediction, but it is standing in a thick fog. It doesn't know the value of any of the patient's medical test results yet. Because it is 'blind' to the specific patient, its prediction is very uncertain, it could be anything from very low to very high. Now, we reveal one test result: blood pressure. Suddenly, the fog clears a little bit. The model is now less uncertain; the range of possible predictions shrinks. Next, we reveal the age of the patient. The fog clears significantly more, and the prediction narrows down further. VARSHAP measures exactly how much 'fog' each feature clears away. If a feature has a high VARSHAP score, it means knowing that feature drastically reduced the model's uncertainty. It was a crucial piece of information. If a feature has a zero VARSHAP score, it means knowing it didn't help clear any fog. So, VARSHAP tells us which features did the heavy lifting in narrowing down the possibilities to reach the final decision.

---

### Author Response · Authors · 2025-11-20
**General comments**

**Impossibility theorem and KernelSHAP being “as good as random”**

Some reviewers confused the Impossibility Theorem (Bilodeau et al. 2024)  with the estimation noise inherent in methods like KernelSHAP (which uses random sampling). Reviewers thought we claimed KernelSHAP is "random" because it uses Monte Carlo sampling and might be inaccurate if n is small. In reality, the Impossibility Theorem applies to the theoretical, exact Shapley value itself, not just its approximation.The theorem states that for any method that satisfies the axioms of Completeness and Linearity (which SHAP does), the resulting attribution depends so heavily on the marginal distribution of the background data that it effectively disconnects from the model's local behavior. Bilodeau shows that you can construct two models that behave in opposite ways locally (e.g., one increases with feature X, one decreases) but produces the exact same SHAP value for feature X because of how the model behaves in distant regions of the data. So, if one uses the SHAP value to decide how to change a feature, one is flipping a coin. The SHAP value mathematically cannot distinguish between the increasing model and the decreasing model. This is why we say standard SHAP is "no better than random guessing" for these tasks, not because the calculation is noisy, but because the definition of the value includes irrelevant global information that obscures the local truth. VARSHAP fixes this by removing the global dependence entirely.

**Local and global explanation vs dependence**

There is significant confusion regarding the terms "local" and "global" in feature attribution. We clarify the definitions used in our paper. In particular, the confusion is caused by the fact that the terms “local” and “global” are used to characterize both explanations and dependence. A standard definition of a “local explanation” is that it explains a single prediction (e.g., why did this applicant get rejected?). A "global explanation" explains the model generally (e.g., which features are most important on average?). Both SHAP and VARSHAP produce "local explanations" in this sense, i.e., they both produce explanations for a single data instance. The main difference between SHAP (and many of its variants) and VARSHAP lies in the dependency. SHAP employs “global dependency” which means that to explain a single instance, SHAP simulates "missing" features by sampling from the entire global dataset (the background distribution). This means the explanation for Patient A is mathematically influenced by Patient Z who is completely different. This is the flaw. VARSHAP defines the "missing" features by sampling only from a tight distribution centered around the instance itself using the Gaussian perturbation. As a result, VARSHAP explains the single instance using only the mathematics of the immediate neighborhood.

---

### Author Response · Authors · 2025-12-03
**Final remarks**

Thank you to Area Chair and all Reviewers for their thoughtful comments and feedback.

During the rebuttal, we clarified the following main concerns (other points are addressed in our detailed responses to individual reviewers):

* **Differences with Sobol indices and ANOVA** - VARSHAP uses local perturbations around the explained instance rather than global distribution, providing an axiomatic solution for local feature attribution distinct from global sensitivity analysis methods.

* **Sign independence property and intuition** - This property prevents the "counterfactual fallacy" where users misinterpret SHAP directionality, and VARSHAP values should be understood as measuring magnitude of influence and uncertainty reduction.

* **Real-world experiments with nonlinear models** - Section 3.3 includes experiments on real-world datasets (Digits images, Wine Quality, Parkinson's Telemonitoring) using nonlinear models (MLPs, CNNs, Decision Trees), not just synthetic toy examples.

* **Benchmark metrics interpretation** - We explained that faithfulness, robustness, and complexity metrics serve complementary purposes in evaluating different aspects of explanation quality.

Additionally, we provided code availability through an anonymized repository and acknowledged the computational complexity trade-off for robustness guarantees.

We believe the discussion has substantially addressed the main concerns raised.

Thank you once more for your time.

Best regards,

The Authors

---

### Meta-Review · Area_Chair_jRbN · 2026-01-06

**Summary:**

This paper addresses the issue of “global dependence” in Shapley-value based methods, identified by Bilodeau et al. (2024), in explainable AI. The authors propose a method (VARSHAP) to alleviate this dependence by modifying the chosen characteristic function to report local variance reduction and then applying standard Shapley aggregation. The reviewers generally find the derivations coherent (X9ZB, s5JJ) and acknowledge the importance of this problem (s5JJ, QD3h). However, the reviewers also question the novelty of the proposed work given other similar takes (such as Sobol/ANOVA/Shapley effects), and several raise concerns about the framing of their contributions and overclaiming, and empirical clarity (X9ZB, s5JJ, k82t, QD3h).

**Reviewer Concerns:**

- Novelty and prior work: reviewer X9ZB raises concerns on overlap with Sobol sensitivity analysis and notes missing citations (Da Veiga 2021). QD3h similarly argues that the issue of distribution dependence is well-studied and requests stronger positioning vs Fumagalli et al. (2025), Sobol indices, and Shapley interactions. Other references not brought up by the reviewers are (Verdinelli et al, 2024). In their rebuttal, the authors argue that VARSHAP is local (i.e. instance-centered perturbations) rather than a global sensitivity analysis (as Anova) and they agree to add Da Veiga (2021) while emphasizing its different scope (responses to X9ZB, QD3h). This concern seems  partially addressed (positioning improved but novelty skepticism remains).

- Too strong claims: both X9ZB and k82t mentioned that the paper is too strong or makes overclaims. The authors did provide an extended clarification distinguishing exact Shapley definition vs Monte Carlo noise (see general comments + replies to X9ZB/k82t).
This issue is somewhat addressed, but it is hard to confirm without a full read of the revised version of the paper by the reviewers, which is not possible at this stage.

- Sign-independence: k82t argues sign-independence makes outcomes harder to interpret and breaks a commonly desired “consistency” intuition; s5JJ further questions whether the paper convincingly ties the modification to the actual understanding of the model interpretability as opposed to improving on some metrics. (see general comments + replies to k82t/s5JJ). I agree with both reviewers and authors here. I do find the problem definition a bit narrow, but I also agree with the authors that the problem addressed by this paper is important if one is to use Shapley-based method to draw any type of conclusions regarding interpretation. Still, reviewer s5JJ remains unconvinced without a rewritten narrative.

- Empirical evaluation and clarity: X9ZB initially thought experiments were mostly toy; QD3h called real-world evaluation unclear and limited; both asked for stronger stress tests. In their rebuttal, the authors pointed to Section 3.3 and the LATEC benchmark on real datasets and nonlinear models (Digits/CNN, Wine Quality/MLP, Parkinson’s, etc.) and clarified the metric categories (faithfulness/robustness/complexity) (see general comments + replies to X9ZB/QD3h). Thus, this comment is addressed, though the reviewers' confusion supports the claim that the paper presentation may not be sufficient.

- Faithfulness results vs KernelSHAP: rev k82t specifically objected that KernelSHAP outperforms VARSHAP on faithfulness metrics. The authors responded that rankings depend on perturbation strategy (random vs black baseline) claiming competitive performance and split wins (authors’ Dec 1 reply). This partially addressed the concern, but then it weakens the preference for VARSHAP in practice.

- Computational complexity: both X9ZB and QD3h asked on the computational complexity of varshap; k82t notes worse complexity than KernelSHAP. This is acknowledged by the authors (about 10× overhead inner-loop variance estimation) and frame as trade-off for robustness guarantees (see responses to QD3h and k82t). This comment is thus addressed, but it hinders the practical applicability of their method.

- Code availability / reproducibility: QD3h criticized missing code; authors later provide an anonymized repository link (response to QD3h). Thus, I consider this addressed.

**Reviewer Scores:**

- rev X9ZB was borderline negative; core issues are novelty/positioning with prior work and experiments. Authors addressed citations (will add Da Veiga) and clarified their real-world experiments, and reframed “solves dependency” as changing the characteristic function per impossibility theorem. I would expect a small upward shift if the reviewer accepts the locality-vs-global distinction and is satisfied that Section 3.3 is genuinely real-world, but some novelty skepticism likely persists.

- rev s5JJ recommended rejection driven primarily by exposition and motivation (“detached from whether it helps users understand”) and paper flow/precision. In discussion, s5JJ explicitly says the replies are compelling but they would need to see a fully revised draft. To me, this indicates that a major re-writing of the paper would be necessary to address their comments.

- rev k82t recommended rejection with concerns about (i) “Shapley concept vs characteristic function” confusion, (ii) sign independence, (iii) faithfulness vs KernelSHAP, and (iv) computational cost. In discussion, k82t explicitly states they raised the score, and would consider raising further if faithfulness concern is addressed; authors responded with a detailed explanation of Table 2’s split behavior depending on perturbation baseline. I would imagine this author increasing their score if it had been possible, perhaps to a 4-5.

- rev QD3h was borderline negative; main issues are general and on the framing of their arguments. lack of code, and limited/unclear real-world evaluation and computational limits. Given that the authors provided strong answer to those comments, including the detail on computational overhead, I envision this reviewer raising their score slightly.

All in all, I think the reviews provided very useful ideas to the authors to improve their presentation. Several related works were brought up. The added clarifications indicate to me that the proposed method has no fundamental flaw. However, it is hard to recommend acceptance given that many of these comments would necessitate from re-writing a more clearly motivated story. For this reason, I am recommending rejection at this stage.

---

### Decision · Program_Chairs · 2026-01-26

Reject